# Go-Explore with a guide: Speeding up search in sparse reward settings with goal-directed intrinsic rewards

## Abstract

Reinforcement Learning (RL) agents have traditionally been very sample-intensive to train, especially in environments with sparse rewards. Seeking inspiration from neuroscience experiments of rats learning the structure of a maze without needing extrinsic rewards, we seek to incorporate additional intrinsic rewards to guide behavior. We propose a potential-based goal-directed intrinsic reward (GDIR), which provides a reward signal regardless of whether the task is achieved, and ensures that learning can always take place. While GDIR may be similar to approaches such as reward shaping in incorporating goal-based rewards, we highlight that GDIR is innate to the agent and hence applicable across a wide range of environments without needing to rely on a properly shaped environment reward. We also note that GDIR is different from curiosity-based intrinsic motivation, which can diminish over time and lead to inefficient exploration. Go-Explore is a well-known state-of-the-art algorithm for sparse reward domains, and we demonstrate that by incorporating GDIR in the "Go" and "Explore" phases, we can improve Go-Explore's performance and enable it to learn faster across multiple environments, for both discrete (2D grid maze environments, Towers of Hanoi, Game of Nim) and continuous (Cart Pole and Mountain Car) state spaces. Furthermore, to consolidate learnt trajectories better, our method also incorporates a novel approach of hippocampal replay to update the values of GDIR and reset state visit and selection counts of states along the successful trajectory. As a benchmark, we also show that our proposed approaches learn significantly faster than traditional extrinsic-reward-based RL algorithms such as Proximal Policy Optimization, TD-learning, and Q-learning.

## 1 Introduction

Recently, LeCun (2022) describes how intrinsic cost modules could motivate an agent's behavior. These intrinsic cost modules can be something like hunger, thirst or goal-seeking that is innate to the agent and cannot be modified. The benefit of having such an immutable cost module is that one's previously learnt values of the state will not be affected by a continually changing model (like the function approximators used in typical Deep Reinforcement Learning (RL)) and the agent can learn efficiently without the need to re-visit past experiences each time the model is changed.

While Silver et al. (2021) states that environment-based extrinsic reward can be enough for learning complex skills and social behaviors, he also admits that it can be sample inefficient. Adding an intrinsic component to this environmental reward can be seen as giving some self-driven intrinsic motivation to learning such skills or behaviors, and can better lead to attaining these competencies with better sample efficiency.

The presence of intrinsic motivation can be seen in neuroscience experiments on rats exploring a maze even without extrinsic food rewards (Fitzgerald et al., 1985; Hughes, 1997). This is not easily explained from just the perspective of extrinsic environmental rewards and suggests that intrinsic motivation does play a huge part in natural behavior of animals, and it could be the critical missing component in modern RL methods.

Motivated by these observations, we seek to find such intrinsic cost/reward functions whereby it is innate to the agent, but is context dependent and can be triggered according to the task at hand. We seek to incorporate this intrinsic cost/reward into RL models and derive the benefits of these intrinsic rewards in driving behavior. Specifically, we model a potential-based goal-directed intrinsic reward (GDIR) which tells an agent how far it is from the goal in order to guide actions.

The proposed GDIR fits naturally in a multi-step task setting, where there can be a planner module that tells us what are the sub-tasks required to achieve the goal. How to derive this planner module is tackled in Hierarchical Reinforcement Learning (Al-Emran, 2015; Pateria et al., 2021). Our proposed GDIR can help to achieve these sub-tasks faster as it provides a learning reward signal regardless of whether the task is achieved.

In contrast to existing work in RL, we do not seek to find the best possible solution for a given environment. We seek instead, to find a *satisficing* solution, whereby the solution is good enough to solve the task. We posit that one pitfall of seeking the optimal solution is that extensive exploration needs to take place, even after solving the environment, in order to ensure that the optimal path is traversed at least once. Hence, we opt for the satisficing route via our novel approach of hippocampal replay, which consolidates successful trajectories and repeats them consistently. This allows our method to learn faster and adapt better to novel environments in real-world systems.

**Our Contributions.** In this paper, we investigate using GDIR in Go-Explore, which is a well-known state-of-the-art algorithm for sparse reward domains. We highlight the following contributions:

1. We propose GDIR, a potential-based goal-directed intrinsic function, which provides a reward signal regardless of whether the task is achieved, and ensures that learning can always take place.
2. We incorporate GDIR into variants of Go-Explore and demonstrate that it enables Go-Explore to learn significantly faster across multiple environments, for both discrete and continuous state spaces.
3. In order to improve consolidation of learnt trajectories, we propose a novel approach of *hippocampal replay* to update the values of GDIR and reset state visit and selection counts of states along the successful trajectory. We demonstrate that hippocampal replay helps an agent remember successful trajectories which solve its current environment and enables it to perform consistently.

## 2 RELATED WORK

**Intrinsic Motivation.** Oudeyer & Kaplan (2009) describes the definition of Intrinsic Motivation in psychology and gives a list of computational approaches to model it. Chentanez et al. (2004) details how to incorporate intrinsic motivation into traditional RL algorithms such as Q-Learning. Schmidhuber (2010) describes a form of intrinsic reward related to discovery of novel, suprising patterns and world model prediction. Baldassarre & Mirolli (2013) details intrinsically motivated learning in natural and artificial systems. Aubret et al. (2019; 2022) gives a general overview of the field of using intrinsic motivation in RL, namely in knowledge acquisition via exploration, empowerment, state space representation, as well as skill learning. Similar to all other prior work on intrinsic motivation, we seek to derive a computational method to incorporate it into a learning system such that it speeds up learning. However, we do not focus on the curiosity-based intrinsic motivation but a goal-directed form of intrinsic rewards, due to reasons explained below.

**Go-Explore.** Go-Explore (Ecoffet et al., 2019) describes two common pitfalls of intrinsic motivation. The first is *detachment*, whereby intrinsic rewards such as curiosity-driven exploration tend to diminish over time, as the frontiers of the exploration space may be explored multiple times but the extrinsic reward may not be obtained. In the end, the frontier may not be attractive anymore and is neglected in subsequent exploration. The second is *derailment*, whereby an agent seeking to return to a previously-explored good state may be hindered from doing so, due to the in-built stochasticity of the algorithm, such as epsilon-greedy exploration (Montague, 1999), state-dependent exploration with action-space noise (Rückstieß et al., 2008), or parameter-space noise (Plappert et al., 2017).

**Extensions of Go-Explore.** First-return-then-explore (Ecoffet et al., 2021) is an extension of Go-Explore using policy-based exploration, which shows the capability of the Go-Explore algorithm to learn on a difficult MuJoCo robotic task which cannot be solved by curiosity-based intrinsic motivation. Yang et al. (2022) details how post-exploration (exploring even after reaching the goal state) can help to expand the agent's state horizon, and even lead to better performance on the same task.

Gallouédec & Dellandréa (2022) describes an extension of Go-Explore which uses dynamically-learnt latent representations.

**Count-based exploration.** While we acknowledge the potential diminishing of curiosity-based rewards affecting search algorithms, a similar approach to balance explore and exploit is still crucial for search problems. As such, we utilize tried and tested count-based exploration as is typically used in Upper Confidence Bounds for Trees (UCT) used in Monte Carlo Tree Search (MCTS) Chaslot et al. (2008). Such an algorithm will guarantee that there will still be exploration in the limit, although the exploration rate will reduce logarithmically. A variant of such a UCT algorithm was used successfully in AlphaGo (Silver et al., 2016), AlphaGo Zero (Silver et al., 2017) and AlphaZero (Silver et al., 2018). In the continuous domain, Bellemare et al. (2016) uses a density-based model as a proxy for count-based exploration methods. In our work, we focus on the discrete case by binning continuous state spaces if necessary. It would be interesting to see if we can also incorporate such a density-based model in future work to deal with continuous spaces directly.

**Goal-directed learning.** We seek to guide the search via a goal-directed approach, much like using a compass to find the way in the forest. One method of learning from binary goal-rewards is Hindsight Experience Replay (HER) (Andrychowicz et al., 2017), where it can be viewed as an implicit curriculum learning method to learn different goal states based on sampling from experienced trajectories. Another way to direct learning towards the goal is to have constraints so as to prevent over-exploration of state space, such as a cost function to encourage RL agents to follow the shortest-path between the start state and the goal state (Sohn et al., 2021). As these methods can be lengthy to learn in practice, we seek to model a potential-based GDIR instead which is equal to 0 when the agent is at the goal, but is of increasing negative value the further the agent is away from the goal. This is very similar to potential-based reward shaping (Grzes, 2017), where the goal states have potential zero, and the non-goal states have non-zero potential. Such a formulation has a theoretical guarantee for optimistic exploration, which is ideal in sparse reward settings. While GDIR may be similar to approaches such as reward shaping in incorporating a goal-based reward, we would like to highlight that GDIR is innate to the agent and hence applicable across a wide range of environments without needing to rely on a properly shaped environment reward.

## 3 METHODS

We detail the agents we use in our paper. There are a total of 7 different agents:

1. **Random Agent.** This agent chooses a valid move randomly. It is the simplest and serves as the worst-case baseline for all other agents.
2. **Q-Learning Agent** We use an online method for Q-Learning updates, with a one-step model-based lookahead and epsilon-greedy policy. Refer to Section A for details.
3. **TD-Learning Agent.** We use an online method for TD-Learning updates, with a one-step model-based lookahead and epsilon-greedy policy. Refer to Section A for details.
4. **PPO Agent.** This agent uses the PPO algorithm (Schulman et al., 2017) implemented on Stable Baselines 3 (Raffin et al., 2019), and is used to benchmark our proposed agents' performance in continuous state space domains due to its superior performance in them.
5. **Go-Explore Agent.** Go-Explore was implemented to be similar to the original paper (Ecoffet et al., 2019; 2021), except that when we select states in the "Go" phase (Refer to Algorithm 2), we do so in a deterministic (rather than probabilistic) manner to help the algorithm learn faster. If GDIR is used, we use it for the "Go" state selection phase in order to gauge how far a state is to the goal state in order to bias exploration towards more promising states, much like how the heuristic function in A* search (Hart et al., 1968) works.
6. **Go-Explore-Count Agent (GDIR).** While Go-Explore uses a random policy for exploration, Go-Explore-Count performs the best action based on a UCT-like selection function. This helps the algorithm explore more promising paths and helps it to solve tougher environments. If GDIR is used, we use it for both the "Go" and "Explore" phase.
7. **Explore-Count Agent (GDIR).** Explore-Count is similar to Go-Explore-Count, except that it does away with the "Go" step and seeks to find the trajectory directly from the start state to the goal state. While this may be inefficient to find the exploration frontier, it allows for exploration without the need to return to the "Go" state which may be impractical for stochastic environments. If GDIR is used, we use it for the "Explore" phase.

**Model-based learning.** We note that we extensively use a model of the environment to make future state predictions, so as to be more efficient in collecting experiences for our agent without the need to traverse all the future states. Such a model of the environment will be easily obtained if one has access to the simulator of the environment, and the exact transition and reward functions need not be known - we only require the use of the model as a black box to make predictions.

**Memory.** The algorithm for Go-Explore variants (Go-Explore, Go-Explore-Count and Explore-Count) is as detailed in Algorithm 1. Of note, we use a similar memory update procedure as in Ecoffet et al. (2019; 2021). For each state, we store in memory: 1) trajectory of actions to reach that state, 2) the number of moves to reach it, 3) the extrinsic reward, 4) the intrinsic reward (unique to our paper), 5) the number of selections of the state in the "Go" phase, and 6) the number of visits to the state in the "Explore" phase. For a new memory state, we will initialize it with the visit and selection counts to 0. For each state we visit, we will update the memory of the next state if the one-step lookahead trajectory has a higher extrinsic reward, or has the same extrinsic reward but shorter trajectory. This is an efficient method to keep the shortest performant trajectory in memory.

**"Go" and "Explore" State selection.** The selection function used for selecting the "Go" state as well as for choosing the best state to "Explore" in all the Go-Explore variants is given by:

$$\alpha ER + \beta IR + \kappa\sqrt{moves} + \gamma\sqrt{numvisited + numselected}, \tag{1}$$

where $ER$ and $IR$ represent extrinsic reward and intrinsic reward respectively, $moves$ represent the minimum number of timesteps taken to reach that state, $numvisited$ represents the number of times the state has been visited, $numselected$ represents the number of times the state has been selected by the "Go" phase of Go-Explore. This will choose states with higher reward and more moves (greater chance of being at the frontier of explored states), while choosing states that have been relatively unvisited and not selected more often. This is similar to the UCT algorithm which decays the exploration term logarithmically so as to encourage greedy action selection in the long run. These attributes are all extracted from memory for the state in consideration.

$\alpha, \beta, \kappa, \gamma$ are hyperparameters to weigh the importance of each term, and are given the respective values 1000, 10, 1, 100. These values can be found using hyperparameter tuning approaches such as grid search or genetic algorithms and can be adapted to the environment in consideration to balance between explore and exploit.

**GDIR.** In the environments considered, we want to make use of a GDIR that is linked to the task completion. We incorporate this GDIR in our Go-Explore variants. This GDIR should range between -1 to 0 so that it can be used in any environment without needing to change the hyperparameters of the selection function. It should be 0 when the task is completed at the goal state, and should be a negative number if the agent is not at the goal state. While it is preferable that the magnitude of the GDIR correlates with how far the agent is away from the goal state for faster solving, it is not required. This GDIR must be easily formulated based on the task conditions alone without any expert domain knowledge and must not give away the solution to the environment - it merely serves as a compass to guide the agent towards more promising areas of the state space. Mathematically, given the current state $c$, and the goal state $g$, GDIR $= -d(c, g)$, where $d(.)$ is an immutable distance function as a proxy to the distance between $c$ and $g$, with output range $[0, 1]$.

**Hippocampal Replay.** Hippocampal replay was performed upon finding a memory with a trajectory that solves the task (i.e. non-zero extrinsic reward) for the Go-Explore variants. This allows us to reset the selection and visit counts of the entire path of states leading to the goal state to 0 as to facilitate exploration along a good path. Furthermore, we let the goal state have an intrinsic reward of 1, and update this path of states to have an intrinsic reward equivalent to the discounted intrinsic reward of the next cell (discount factor $\gamma = 0.99$ in our case), so to encourage choosing back this sequence of moves during future exploration. Unlike Mnih et al. (2013) and Schaul et al. (2015), we do not randomly sample or priority sample the buffer, but instead just perform one forward play (pre-play) in order to do memory retrieval and get the entire list of states along the trajectory, and a reverse play (replay) so as to update the intrinsic rewards and reset state selection and visit counts. This is akin to biological pre-play and replay patterns shown in hippocampal replay (Joo & Frank, 2018) in mice. It is more efficient, as in one hippocampal replay step, all the states can be updated with the correct intrinsic values, as opposed to updating just one transition sampled from the buffer. The algorithm for hippocampal replay is detailed in Algorithm 3 and illustrated in Fig. 1.

# 4 EXPERIMENTS

We compare the agents' performance in two types of environments. The first set of environments are those with discrete state values and discrete actions which are typically used in planning and problem solving. The second set of environments are those with continuous state values and discrete actions which are usually used for robotic control. We intentionally choose sparse reward environments, or modify the environments so as to make the rewards sparse. This will pose difficulty for traditional RL algorithms, and hence enable us to explore the limits of our proposed Go-Explore variants.

## 4.1 DISCRETE STATE ENVIRONMENTS

In total, we test our agents on four environments with a discrete state space and action space. They are namely: Unwalled Maze, Walled Maze, Towers of Hanoi, Nim. The environment specifications are detailed in Table 6. The unwalled and walled mazes are designed to test easy and difficult spatial navigation respectively. Towers of Hanoi is a more difficult search problem with a specific sequence of moves to follow, though there is room for a few redundant moves to achieve the goal since we give the agents a move limit which is 4 times more than the optimal number of moves needed. Nim, repackaged as a single player environment with perfect play opponent (See Section D for strategy for perfect play), is the hardest search problem whereby we only have one possible option per move in order to win the perfect play opponent. These environments are proposed in order to test the efficiency of the Go-Explore variants on problems of increasing difficulty. To provide a baseline, we also compare the performance of the Go-Explore variants against the Random, Q-Learning and TD-Learning agents.

## 4.2 CONTINUOUS STATE ENVIRONMENTS

In total, we test our agents on two environments with a continuous state space and discrete action space. They are namely: Cart Pole and Mountain Car. These environments are part of the Classical Control environments on OpenAI Gym (Brockman et al., 2016). Rather than use the original reward function, we make the problem harder by using only a sparse reward function for the environments, where a reward of 1 is only given if the agent solves the problem, and 0 otherwise. The environment specifications are detailed in Table 7. Cart Pole is a problem where we need to continue to maintain the initial state of a balanced pole which we were in, while Mountain Car is a challenging problem with a goal state that requires going further away from it to reach it. To provide a baseline, we also compare the performance of the Go-Explore variants against the competitive PPO agent.

**State Space Abstraction.** Due to the limitation of the memory approach we are using for Go-Explore, we cannot deal with a continuous state space which has infinite possibilities. As such, we drastically simplify the state space by binning the continuous state space. For our bins, we choose the size of $0.01$ for position values, and $0.001$ for velocity values, which we found to give good performance. How to choose the right bin based on the environment dynamically can be a topic of future research. As some values can also span the range of $-\infty$ to $\infty$, we have an upper and lower limit for binning to prevent infinitely many states. We set this upper/lower limit to be $\pm 42$ for Cart Pole and $\pm 50$ for Mountain Car in order to match the maximum allowable pole angle of $\pm 0.418$ and goal state car x-position of $0.5$ respectively. Mathematically, we describe the binning process as such: given bin size $m$ and upper/lower limit of bin $u$, we discretize the original continuous state value $v$ to a discrete value $b = \min(\max(\lfloor \frac{v}{m} \rfloor, -u), u)$.

## 4.3 GOAL DIRECTED INTRINSIC REWARD

We use different GDIR functions across our various environments, as detailed in Tables 6 and 7.

**Discrete Environments.** For the maze environments, the GDIR is proportional to the Manhattan distance of agent's position to the door position, which only tells the agent the shortest distance to the goal if the walls are not present - the agent still has to figure out how to navigate these walls. For Towers of Hanoi, the GDIR is proportional to the number of disks matching the goal state, which similarly does not tell the agent how to achieve the goal state but merely serves as a guide. For Nim, the GDIR is proportional to the number of matchsticks remaining after each move, which encourages the agent to reach 0 matchsticks but does not tell the agent how to.

**Continuous Environments.** For Cart Pole, the GDIR is proportional to the absolute value of the cart position $x_c$ and the absolute value of the pole angle $p_\theta$, since it is given that the environment will terminate early if $x_c < -2.4$ or $x_c > 2.4$ or $p_\theta > 0.2095$ rad or $p_\theta < 0.2095$ rad. Incorporating this GDIR is fair because a human who plays the game will already know of these conditions, whereas the reward structure in the environment will be sparse and the agent will have to figure out for itself how not to terminate the environment from trial and error. For Mountain Car, the GDIR is proportional to the distance away from the end state of car x-position $x = 0.5$. This similarly tells the agent that the goal is to move towards $x = 0.5$, which is also known by a human, but does not give away the solution that the car needs to move left and right along the hill valley in order to increase velocity to reach the top of the hill.

### 4.4 EVALUATION DETAILS

For each environment, the agent will act and learn in the environment for a total of 100 runs (episodes). At the beginning of evaluation, we first initialize $memory$ to be an empty dictionary to be updated across the 100 runs. At the start of each run, we perform the "Go" phase to select a state to start from (only for Go-Explore variants). Thereafter, for each time step till termination, we query the agent for the next move. At the end of each run, we will perform hippocampal replay to consolidate a good trajectory if there are any (only for Go-Explore variants). The entire procedure is described in in Algorithm 4. For some experiments such as those involving neural networks or random variables, we perform 10 trials of 100 runs, and report the average across the 10 trials. Specifically, we are interested in the run where the agent *first solves* the environment as that is related to how adaptable it is to a new environment, as well as the *minimum number of steps* in any found solution trajectory as that is related to how efficient the algorithm is.

## 5 RESULTS

### 5.1 DISCRETE STATE ENVIRONMENTS

**Overview.** We compare the run where the first solve occurs (adaptability) and the minimum solve time steps (efficiency) of any successful run out of 100 runs across all algorithms in Tables 1 and 2 respectively. We consider Unwalled Maze and Walled Maze with dimensions 10x10, 20x20 and 100x100, Towers of Hanoi with 3 and 7 disks, and Nim with 11, 21 and 1001 starting matchsticks. The full experimental results are available at Sections E, F, G and H.

**First Solve.** Overall, Explore-Count GDIR performs the best, with a first solve run of 1 or 2 for most environments, except for Towers of Hanoi with 7 disks where it needs 10. Go-Explore-Count GDIR also does very well, except that it does not solve Towers of Hanoi with 7 disks. Even without GDIR, Explore-Count and Go-Explore-Count still perform well, but may require slightly more runs for first solve. Random, TD-Learning, Q-Learning and Go-Explore do not perform well for complex environments like Walled Maze, Towers of Hanoi, Nim, often failing to solve the environments.

**Minimum Steps.** In terms of minimum steps, Explore-Count GDIR performs the best for Walled Maze and Nim. For simple environments like Unwalled Maze, TD-Learning performs the best. This highlights that traditional RL algorithms perform efficiently if the state space is small, but Explore-Count performs quite well too even in complex environments. Go variants (Go-Explore, Go-Explore-Count) do not perform very efficiently, often having higher minimum steps.

**Problems with jumping to the "Go" state.** Go-Explore does not perform well and often does not solve the environments. This may be because deterministically selecting the "Go" state and prioritizing states which are unvisited often means that states with longer trajectories are selected. Perhaps if we had followed the probabilistic selection employed in Ecoffet et al. (2019), we could have solved more environments at the cost of a longer training time. Furthermore, unlike in Ecoffet et al. (2019), the environments we considered all have a step limit, hence picking a "Go" state with a long trajectory may mean that there is very little room left to explore. This trait of Go-Explore needing long trajectories can also be seen in Ecoffet et al. (2019) where the agent in Montezuma's Revenge exceeded the maximum time steps of the default OpenAI gym environment. Perhaps Go-Explore methods will be more viable if we do not limit the maximum timesteps. That said, with a more efficient exploration policy using count-based methods, Go-Explore-Count mitigates the issues of jumping to the "Go" state and does quite well for the environments.

Table 1: First Solve Run for Discrete State Environments (lower is better, '-' means unsolved)

| Agent | Unwalled | | | Walled | | | Hanoi | | Nim | | |
|---|---|---|---|---|---|---|---|---|---|---|---|
| | **10** | **20** | **100** | **10** | **20** | **100** | **3** | **7** | **11** | **21** | **1001** |
| Random | 8 | - | 15 | - | - | - | 8 | - | 71 | - | - |
| TD-Learning (Train) | 21 | 7 | 15 | - | - | - | 4 | - | 27 | - | - |
| TD-Learning (Test) | **1** | **1** | **1** | - | - | - | **1** | - | **1** | - | - |
| Q-Learning (Train) | 21 | 7 | 15 | - | - | - | 4 | - | 27 | - | - |
| Q-Learning (Test) | **1** | - | - | - | - | - | **1** | - | **1** | - | - |
| Go-Explore | - | - | - | - | - | - | 33 | - | 7 | 5 | - |
| Go-Explore GDIR | 12 | - | - | 75 | - | - | 33 | - | 6 | 5 | - |
| Go-Explore-Count | **1** | 2 | **1** | **1** | **1** | **1** | **1** | - | 2 | **2** | **2** |
| Go-Explore-Count GDIR | **1** | **1** | **1** | **1** | **1** | **1** | **1** | - | **1** | **2** | **2** |
| Explore-Count | **1** | 2 | **1** | **1** | **1** | **1** | **1** | 17 | 2 | 3 | 3 |
| Explore-Count GDIR | **1** | **1** | **1** | **1** | **1** | **1** | **1** | 10 | **1** | **2** | **2** |

Table 2: Min Steps to solve for Discrete State Environments (lower is better, '-' means unsolved)

| Agent | Unwalled | | | Walled | | | Hanoi | | Nim | | |
|---|---|---|---|---|---|---|---|---|---|---|---|
| | **10** | **20** | **100** | **10** | **20** | **100** | **3** | **7** | **11** | **21** | **1001** |
| Random | 54 | - | 9980 | - | - | - | 17 | - | **3** | - | - |
| TD-Learning (Train) | 44 | 264 | 5544 | - | - | - | 13 | - | **3** | - | - |
| TD-Learning (Test) | **18** | **38** | **198** | - | - | - | **7** | - | 3 | - | - |
| Q-Learning (Train) | 44 | 264 | 5544 | - | - | - | 13 | - | **3** | - | - |
| Q-Learning (Test) | **18** | - | - | - | - | - | **7** | - | **3** | - | - |
| Go-Explore | - | - | - | - | - | - | 13 | - | **3** | 6 | - |
| Go-Explore GDIR | 62 | - | - | 92 | - | - | 13 | - | **3** | 6 | - |
| Go-Explore-Count | 62 | 356 | 7810 | 64 | 218 | 4912 | 9 | - | **3** | 6 | **251** |
| Go-Explore-Count GDIR | 20 | 42 | 230 | 60 | 220 | 4042 | 9 | - | **3** | 6 | **251** |
| Explore-Count | 22 | 76 | 482 | 42 | 138 | 3160 | 9 | 378 | **3** | 6 | **251** |
| Explore-Count GDIR | 20 | 42 | 222 | **34** | **82** | **1424** | 9 | 327 | **3** | 6 | **251** |

## 5.2 CONTINUOUS STATE ENVIRONMENTS

**Overview.** We compare the run where the first solve occurs (adaptability) across all algorithms in Table 3. We consider Cart Pole with the goal state of surviving for 50, 100 and 175 time frames. As Cart Pole has a high chance of encountering the same state (usually the equilibrium state where pole is balanced), which can negatively affect count-based algorithms by artificially inflating the visit counts, we also consider encoding repeated states as a distinct new state in memory (variants with 'R' in them). For Mountain Car, we also consider Mountain Car Repeat with selected actions repeated 10 times, and car velocity binned by 0.01 instead of 0.001 accordingly. The full experimental results are available at Sections I and J.

**First Solve Run.** Overall, Go-Explore-Count GDIR performs the best for Cart Pole 50 with a first solve run of 3, while Go-Explore GDIR performs the best for Cart Pole 175 with a first solve run of 20. They both perform equally well for Cart Pole 100 with a first solve run of 6. Explore-Count GDIR performs the best for Mountain Car with a first solve run of 7. These all learn faster and are more adaptable than PPO. Contrasting with the results from the discrete environments, this highlights that the "Go" step is useful if solving the environment is linked to long trajectories.

**Repeated States/Actions.** For Cart Pole 175 and Mountain Car, the non-GDIR methods fail to solve it, while PPO still manages to, indicating that for extended search problems which involve similar actions, a trained neural network might work better to repeat a performant sequence of actions. That said, for Cart Pole 175R, Go-Explore and Go-Explore-Count and their GDIR variants have a lower first solve run than PPO, which highlights distinct repeated states could be helpful for tasks with repeated states. For Mountain Car Repeat, Go-Explore GDIR has a lower first solve run than PPO.

Table 3: First Solve Run for Continuous State Environments (lower is better, '-' means unsolved). PPO results are averaged over 10 trials (refer to Tables 36 and 39 for details).

| Agent | Cart Pole | | | | | | Mountain Car | |
|---|---|---|---|---|---|---|---|---|
| | **50** | **50R** | **100** | **100R** | **175** | **175R** | **Normal** | **Repeat** |
| PPO (Discretized State) | 3.7 | 3.7 | 39.7 | 39.7 | 62.3 | 62.3 | 43.1 | 3 |
| PPO (Continuous State) | 6.6 | 6.6 | 55.3 | 55.3 | 60 | 60 | 92.6 | 12.5 |
| Go-Explore | 4 | 4 | 22 | 22 | - | 51 | - | 38 |
| Go-Explore GDIR | 4 | 4 | **6** | 6 | **20** | **21** | - | **2** |
| Go-Explore-Count | **3** | **2** | 27 | 5 | - | 26 | - | - |
| Go-Explore-Count GDIR | **3** | **2** | **6** | 5 | 28 | 40 | - | 72 |
| Explore-Count | **3** | 4 | 56 | **4** | - | 30 | - | 10 |
| Explore-Count GDIR | 4 | 4 | 69 | **4** | - | 75 | **7** | 3 |

Table 4: Performance of Explore-Count GDIR on Walled Maze 100x100 with different intrinsic rewards. For random instrinsic rewards in a given range $[.,0]$, results are averaged over 10 trials.

| Attribute | Intrinsic Reward | | | | | | |
|---|---|---|---|---|---|---|---|
| | **Manhattan** | **0** | **[-1,0]** | **[-5,0]** | **[-10,0]** | **[-20,0]** | **[-50,0]** |
| First Solve | **1** | **1** | 1.7 | 1.9 | 1.7 | 10.5 | 96.2 |
| Solve Rate | **100/100** | **100/100** | 99.3/100 | 99.1/100 | 99.3/100 | 90.5/100 | 4/100 |
| Memory Size | **5069** | 7552 | 7725.6 | 7991.7 | 7629.8 | 9628.9 | 9692.5 |
| Min Steps | 1424 | 3160 | 655 | **557** | 762 | 730.4 | 561 |

**Improving PPO performance.** PPO solves the environments even with a continuous state space, which highlights the generalization capabilities of a neural network. With discretized binning of the continuous states, PPO performs even better, highlighting the usefulness of abstracting state space.

### 5.3 ABLATION STUDY

**What if intrinsic rewards are wrongly calibrated?** We seek to test what would happen if we use uninformed intrinsic rewards of increasing magnitude in the Walled Maze 100x100 environment. From Table 4 (detailed results in Section F.3), it can be seen that as the magnitude of uninformed (random) intrinsic reward increases, the overall solve rate decreases, first solve run increases, and first solve memory size increases, indicating more exploration of the environment. This increased exploration may lead to a lower minimum step, but at the cost of slower adaptation or even failing to solve the environment. Overall, we can see that GDIR can help solve the environment faster if calibrated correctly, and is important for fast adaptability to a new environment.

**What if we remove hippocampal replay?** We evaluate Go-Explore variants on the Walled Maze 100x100 environment with and without hippocampal replay. From Table 5, hippocampal replay does not affect first solve run and memory size, since hippocampal replay only happens after a successful first solve. After the first solve, having no hippocampal replay causes the agent to explore, which leads to lower solve rate and higher maximum step. While increased exploration may allow the agent to attain a lower minimum step, hippocampal replay can consolidate a good trajectory and repeat it consistently. Overall, hippocampal replay is very important for consistent performance and helps an agent remember successful trajectories which solve its current environment.

### 6 DISCUSSION

**Advantage of GDIR.** Overall, we can see that GDIR enables the Go-Explore variants to learn signficantly faster. The main advantage is fast adaptation (low first solve run) to a new environment, even in continuous state space environments. GDIR works much like a compass to guide initial actions that can increase the chance of solving an environment.

Table 5: Performance results for Walled Maze 100x100 with and without Hippocampal Replay. '*' represents without hippocampal replay. Bolded text represents better performance.

| Overall | | First Solve | | Steps to Solve | | |
|---|---|---|---|---|---|---|
| Agent | Solve Rate | Run | Memory size | Avg | Min | Max |
| Go-Explore-Count | 100/100 | 1 | 7552 | **4912.0** | 4912.0 | **4912.0** |
| Go-Explore-Count* | 100/100 | 1 | 7552 | 4918.2 | **4718.0** | 6362.0 |
| Go-Explore-Count GDIR | 100/100 | 1 | 5069 | 4042.0 | 4042.0 | **4042.0** |
| Go-Explore-Count GDIR* | 100/100 | 1 | 5069 | **3891.4** | **3676.0** | 5326.0 |
| Explore-Count | **100/100** | 1 | 7552 | **3177.5** | **3160.0** | **4912.0** |
| Explore-Count* | 32/100 | 1 | 7552 | 7376.2 | 3674.0 | 9954.0 |
| Explore-Count GDIR | **100/100** | 1 | 5069 | **1450.2** | 1424.0 | **4042.0** |
| Explore-Count GDIR* | 34/100 | 1 | 5069 | 4611.6 | **1410.0** | 9818.0 |

**Advantage of Count-based Go-Explore and Hippocampal Replay.** For most of the environments, Go-Explore-Count and Explore-Count learns significantly faster than traditional RL algorithms such as TD-Learning and Q-Learning, even without GDIR. With GDIR, it learns significantly faster than PPO in all continuous environments considered. Hippocampal replay is also beneficial as it enables consolidation of good trajectories for higher chance of repetition in the future. This makes it suitable for use in a real-life robot, where optimization is a good-to-have (unlike in game-based RL environments such as Chess or Go), and fast adaptation to new situations is key for performance.

**Satisficing Solution.** Although the Go-Explore variants do not seek to explore further upon solving the environment, by nature of the fact that we update our memory with the shortest possible sequence of actions to reach that state, the agent's performance can be quite close to optimal.

**Robustification.** The approach presented in our paper is only the first step of Go-Explore to gather successful trajectories. In order to generalize to stochastic, continuous settings, we can then perform imitation learning on trajectories obtained into a neural network (Ecoffet et al., 2019; 2021).

## 7 CONCLUSION AND FUTURE WORK

The field of RL has come a long way, and we now seek to focus on sample-efficient methods for real-world adapation for our RL agents, which we propose to do using Go-Explore variants with GDIR and hippocampal replay. In fact, the proposed GDIR and hippocampal replay methods are not just limited to Go-Explore and can even be used to improve other RL architectures. We hope that this paper, inspired from the vision of Yann LeCun (LeCun, 2022) and other pioneers of incorporating intrinsic rewards to RL, would help to pave the way for more efforts focusing on this domain.

**Incorporating Neural Networks for Generalization.** The proposed count-based approach is most effective for discrete state and action spaces. For continuous state and action spaces, we could utilize neural networks as the exploration policy, similar to policy-based Go-Explore (Ecoffet et al., 2021), while incorporating density models as a proxy for count-based models (Bellemare et al., 2016). Neural networks can also help improve our method by providing another means of abstracting similar states together. It can even be used for world model prediction, perhaps using an architecture similar to Decision Transformers (Chen et al., 2021). We intend to explore this in future work.

**Configurator Module.** Another line of future work will be to design a configurator-like module, which can select the relevant GDIR modules to use for a given environment. This can be done via an attention-based mechanism like that in a Transformer (Vaswani et al., 2017), which uses the relevant reward functions based on similarity of its intended use case to the environment.

**Multi-Agent Learning.** We envision a scenario where rather than just having one agent learning by itself, we can have multiple agents with different hyperparameters learning from each other. This can help increase the chance of solving the environment, and knowledge sharing can take place via hippocampal replay of successful trajectories across all agents, thus expediting the learning process.

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

# A  Q-LEARNING AND TD-LEARNING AGENTS

## A.1  Q-LEARNING

Consider a Markov Decision Process (MDP) defined by the tuple $(S, A, R, P)$, where $S$ represents the set of states, $A$ represents the set of actions, $R$ represents the reward function between transitions of states, given by the function $R(s_t, a_t, s_{t+1})$, $P$ represents the transition probability of going from one state to another, given by the function $P(s_{t+1}|s_t, a_t)$, and $t$ is the timestep.

We will perform an online learning for the Q-functions, where we only update the state-action values that we visit for efficiency. For every state transition $(s_t, a_t, s_{t+1})$, we calculate the TD-error:

$$\delta_t = r_t + \gamma \max_{a \in A} Q(s_{t+1}, a) - Q(s_t, a_t). \tag{2}$$

Here, in order to calculate all future states $s_{t+1}$, we will treat the model as in-built to the agent and use it to calculate all future values $Q(s_{t+1}, a)$. Doing so will speed up convergence as we can evaluate all possible futures in one update.

Thereafter, we perform the Q-learning update:

$$Q(s_t, a_t) \leftarrow Q(s_t, a_t) + \alpha \delta_t. \tag{3}$$

We use an $\epsilon$-greedy behavior policy, where we take a random action a fraction $\epsilon$ of the time, and the greedy action $a^* = \max_{a \in A} Q(s_{t+1}, a)$ a fraction $1 - \epsilon$ of the time. For our agent, the training phase uses $\epsilon = 1$ to explore the state space, and the testing phase uses $\epsilon = 0$ to greedily select actions.

## A.2  TD-LEARNING AGENT

The TD-Learning agent we consider is simply the online Q-Learning agent applied to state values instead of state-action values. This is slightly different from the model-free TD-$\lambda$ approaches (Sutton & Barto, 2018) in that it is actually a model-based approach where we will need a model to compute all possible next states after taking an action from the current state. The advantage of doing such an approach is that we can be more efficient at performing our value updates by comparing against all future possible states at the same time.

Similar to the Q-learning agent, for every state transition $(s_t, a_t, s_{t+1})$, we calculate the TD error:

$$\delta_t = r_t + \gamma \max_{a \in A, a:s_t \to s_{t+1}} V(s_{t+1}) - V(s_t). \tag{4}$$

Thereafter, we perform the value update:

$$V(s_t) \leftarrow V(s_t) + \alpha \delta_t. \tag{5}$$

Similar to Q-Learning, we use an $\epsilon$-greedy behavior policy, where we take a random action a fraction $\epsilon$ of the time, and the greedy action $a^* = \max_{a \in A, a:s_t \to s_{t+1}} V(s_{t+1})$ a fraction $1 - \epsilon$ of the time. For our agent, the training phase uses $\epsilon = 1$ to explore the state space, and the testing phase uses $\epsilon = 0$ to greedily select actions.

For both Q-learning and TD-learning agents, we choose hyperparameters $\gamma = 0.99$ and $\alpha = 1$

# B   ALGORITHMS

We detail the algorithms used for our agents in Algorithms 1, 2, 3 and 4.

---

**Algorithm 1** Go-Explore-Variants "Explore" phase

---

1: **procedure** GO-EXPLORE-VARIANTS($env$, $action\_trajectory$, $replay$, intrinsic_function):
2:     $intrinsic\_reward$ = intrinsic_function($env$)                    ▷ 0 if no intrinsic function
3:     $curmem$ = $memory[env.state]$                    ▷ Add $env.state$ to memory if not present
4:     **if** $env.done$ **then**          ▷ Make the intrinsic value of successful trajectory end state be 1
5:         **if** $env.reward > 0$ **then** $curmem['intrinsic'] = env.reward$
6:         **return**
7:     **if** not $replay$ **then**                    ▷ Increment state visit count by 1
8:         $curmem.numvisited = curmem.numvisited + 1$
9:     **else if** $replay$ **then**          ▷ If replay, then reset counts to prioritize choosing the state again
10:         $curmem.numvisited = 0$
11:         $curmem.numselected = 0$
12:     **for** $move$ in $env.validmoves$ **do**
13:         $newenv$ = copy($env$)                    ▷ Replicate current environment
14:         $newenv = newenv.step(move)$          ▷ Simulate move in the replicated environment
15:         $next\_intrinsic\_reward$ = intrinsic_function($newenv$)          ▷ 0 if no intrinsic function
16:         $nextmem = memory[newenv.state]$    ▷ Add $newenv.state$ to memory if not present
17:         Update $nextmem$ with attributes of $newenv$ if there is higher extrinsic reward at next state compared to memory, or a shorter path exists with same extrinsic reward, and reset $numvisited$ and $numselected$ to be 0
18:         $total\_reward$ = selection_function($nextmem$)
19:     $best\_intrinsic\_reward$ = highest $next\_intrinsic\_reward$ across all $move$
20:     $curmem.intrinsic\_reward = \gamma \cdot best\_intrinsic\_reward$                    ▷ $\gamma$ set to 0.99
21:     **if** GO-EXPLORE **then**
22:         return random move in $env.validmoves$                    ▷ Random Policy
23:     **else if** GO-EXPLORE-COUNT or EXPLORE-COUNT **then**
24:         return $move$ with highest $total\_reward$                    ▷ Greedy Policy

---

**Algorithm 2** Choosing State for "Go"

---

1: **procedure** CHOOSESTATE($env$):
2:     **for** $eachmemory$ in $memory$ **do** $total\_reward$ = selection_function($eachmemory$)
3:     $bestmem = eachmemory$ with highest $total\_reward$
4:     **for** $move$ in $bestmem.action\_trajectory$ **do** $env.step(move)$
5:     $bestmem.numselected = bestmem.numselected + 1$    ▷ increment selection count by 1
6:     return $bestmem.action\_trajectory$, $env$

---

**Algorithm 3** Hippocampal Replay

---

1: **procedure** HIPPOCAMPALREPLAY($env$, $action\_trajectory$, intrinsic_function):
2:     $env\_action\_list = []$
3:     $action\_history = []$
4:     **for** $move$ in $action\_trajectory$ **do**                    ▷ Pre-play
5:         $env\_action\_list.append($copy($env$), $action\_history$)
6:         $env.step(move)$
7:         $action\_history.append(move)$
8:     **for** $env$, $action\_history$ in reversed($env\_action\_list$) **do**                    ▷ Replay
9:         Go-Explore-Variants(copy($env$), $action\_history$, $replay$ = True, intrinsic_function)

---

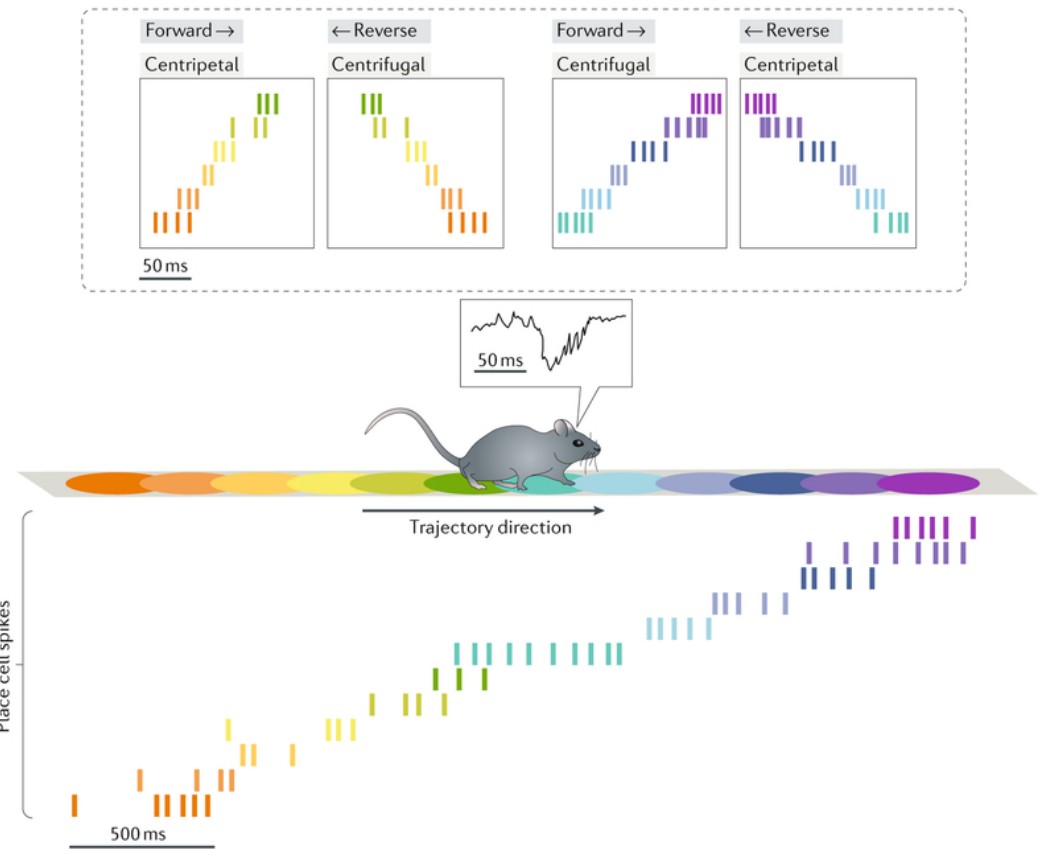

Figure 1: Hippocampal replay in mice, which showcases forward play (pre-play) and reverse play (replay), which are involved in memory retrieval and consolidation for processes such as decision-making. Extracted from Fig. 2 of Joo & Frank (2018). There is replay occuring for both 1) past visited states and 2) future unvisited states. We only focus on replay for past visited states, and use these insights in designing Algorithm 3 for consolidating learnt experiences.

---

**Algorithm 4** Evaluation

---

1: **procedure** EVALUATION($env$, $agent$, $numruns$, $maxsteps$, intrinsic_function):
2:     global $memory = []$                                              ▷ Global variable
3:     **for** $i$ in range($numruns$) **do**
4:         $env.reset()$                                      ▷ Reset to original state of environment
5:         $action\_trajectory = []$
6:         **if** agent is Go-Explore or agent is Go-Explore-Count **then**
7:             $action\_trajectory, env$ = ChooseState($env$)                      ▷ "Go" phase
8:         **while** $env.done$ and $env.numsteps < maxsteps$ **do**             ▷ "Explore" phase
9:             $move = agent(env, action\_trajectory, replay$ = False, intrinsic_function)
10:            $env.step(move)$
11:            $action\_trajectory.append(move)$
12:        $agent(env, action\_trajectory, replay$ = False, intrinsic_function)
13:        **if** agent is a Go-Explore-Variant and $reward$ 1 in $memory$ **then**            ▷ Replay
14:            $bestmem = memory$ with $reward$ 1 ▷ Choose a memory which has solved the task
15:            $env.reset()$                                   ▷ Reset to original state of environment
16:            HippocampalReplay(copy($env$), $bestmem.action\_trajectory$, intrinsic_function))

---

Table 6: List of discrete state environments and their specifications

| | Unwalled Maze | Walled Maze | Towers of Hanoi | Nim |
|---|---|---|---|---|
| **Image** | ``X.#...#...``
``#..#......``
``#.........``
``........#.``
``..........``
``..........``
``.........#``
``.....#....``
``#.....#...``
``.........D`` | ``X...#.....``
``....#...#.``
``....#...#.``
``....#...#.``
``....#...#.``
``##.###.##.``
``....#...#.``
``........#.``
``....#...#.``
``....#...#D`` |  |  |
| **State Space** | Agent position (x and y axis) | Agent position (x and y axis) | Disk positions on the respective poles | Number of matchsticks remaining |
| **Action Space** | Up, Down, Left, Right to an available open area (.). No wraparound. | Up, Down, Left, Right to an available open area (.). No wraparound. | Move top disk from one pole to another (must be smallest on destination pole). | Remove 1-$K$ matchstick(s) |
| **Start State** | • Maze of size $Height \times Width$
• Agent (X) at top left, Door (D) at bottom right
• 10% of grid cells generated as bricks (#) which prohibit agent movement | • Maze of size $Height \times Width$
• Agent (X) at top left, Door (D) at bottom right
• Bricks (#) form 4 chambers plus one final narrow pathway to door, which make solving difficult | • 3 poles labelled A, B, C
• $N$ disks stacked smallest to largest from top to bottom on Pole A
• Poles B and C are initially empty | • $N$ matchsticks in a pile
• 2 player game turn-based game, agent will start first
• Opponent plays perfectly, in the event of a losing state, agent just picks 1 matchstick |
| **Goal State** | Agent reaches door position | Agent reaches door position | $N$ disks stacked smallest to largest from top to bottom on Pole C | 0: Be the player to remove the last matchstick |
| **Max Timestep** | $Height * Width$ | $Height * Width$ | $2^{N+2}$ | $N$ |
| **Extrinsic Reward** | 1 if goal state reached before max timestep, 0 otherwise | | | |
| **Intrinsic Reward** | $-($Manhattan Distance of agent to door position$)/(Height + Width - 2)$ | $-($Manhattan Distance of agent to door position$)/(Height + Width - 2)$ | (Number of disks matching goal state $-N)/N$ | $-$Matchsticks remaining after move$/N$ |

Table 7: List of continuous state environments and their specifications

| | **Cart Pole** | **Mountain Car** |
|---|---|---|
| **Image** |  |  |
| **State Space** | • Cart x-position $x_c$: -4.8 to 4.8
• Cart velocity: $-inf$ to $inf$
• Pole angle $p_\theta$: -0.418 to 0.418 rad
• Pole ang velocity $p_v$: $-inf$ to $inf$ | • Car x-position $x$: -inf to inf
• Car velocity $v$: -inf to inf |
| **Action Space** | Push cart left or right | Accelerate to the left, don't accelerate, accelerate to the right |
| **Start State** | All state values at 0 | $x$ at $-0.5$, $v$ at $0$ |
| **Goal State** | Not reach early termination for $N$ timesteps | Car at 0.5 x-position |
| **Max Timestep** | 200 | 200 |
| **Early Termination** | • $p_\theta > 0.2095$ rad or $< 0.2095$ rad
• $x_c < -2.4$ or $> 2.4$ | • Nil |
| **Extrinsic Reward** | 1 if goal state reached before max timestep or early termination, 0 otherwise | |
| **Intrinsic Reward** | $-\frac{0.5|x_c|}{2.4} - \frac{0.5|p_\theta|}{0.2095}$ | $0.5 - x$ |

# D  Strategy for Nim

## D.1  Overview

Nim is a game with $N$ matchsticks in a pile, and a move which can range from $1 - K$ matchsticks. It is an alternating two-player turn-based game. The player who removes the last matchstick wins.

## D.2  Working backwards from win state

Given that the win state is to remove the last matchstick, we then see that if one receives a state of $\leq K$ matchsticks at the start of one's turn, one can win by simply removing all the matchsticks. Hence, the losing state will then be receiving a state of $K + 1$ matchsticks, because for any possible move one makes, one would necessarily let the opponent have a state of $\leq K$ matchsticks and hence lose. Hence, the goal is to then present the opponent with a state of $K + 1$ matchsticks at some point of time in the game in order to win.

## D.3  A strategy for all opponent responses

Regardless of how the opponent responds (choosing a move between $1 - K$ matchsticks), it is possible to maintain a sum of $K + 1$ between both turns the opponent and oneself. This is easy to verify, because if the opponent chooses $i$, one can respond with $K + 1 - i$. This will always be a valid move since $1 \leq i \leq K$ and hence $1 \leq K + 1 - i \leq K$. Hence, knowing that presenting a state of $K + 1$ matchsticks to the opponent is a win state, and that having a interval of $K + 1$ between the opponent's move and one's move is always possible, one would hence want to present a state of $K + 1, 2(K + 1), 3(K + 1), ...., p(K + 1)$ to the opponent, where $p \in \mathbb{N}N$.

**Winning State.** In short, since the winning strategy is always to present a state which is a multiple of $(K + 1)$ to your opponent, we can then determine the winning play at any point of time with certainty. We simply need to take a move which presents a state of a multiple of $K + 1$ to the opponent. In short, we need to take $X \% (K+1)$ matchsticks, where $X$ is the number of matchsticks you are presented with originally, and $\%$ is the modulo operator.

**Losing State.** However, this number of $X \% (K + 1)$ may be 0, which is not a valid move. If that is the case, the player who is presented with this state will always lose if the opponent plays perfectly. In this case, our perfect play agent will always take 1 matchstick (though it is also possible to randomly select between all possible moves).

## D.4  The winning algorithm

Having seen the logic behind the perfect play agent, we detail the algorithm needed to play Nim perfectly in Algorithm 5 below. Note that $X$ is the number of matchsticks presented to the agent.

---
**Algorithm 5** Perfect Play Agent for Nim

---
1: **procedure** NimPerfectPlay($X$, $K$):
2:     **if** $X \% (K + 1) > 0$ **then** return $X \% (K + 1)$                    ▷ Give opponent the losing state
3:     **else** return 1                    ▷ This is a losing state, so return any valid move

---

# E FULL RESULTS FOR UNWALLED MAZE

## E.1 WITH HIPPOCAMPAL REPLAY

Tables 8, 9 and 10 detail the results for unwalled maze with hippocampal replay. A '-' represents unsolved after 100 runs, and bolded text means best result across all agents. TD-Learning, Go-Explore-Count and Explore-Count and their GDIR variants perform the best.

Table 8: Performance results for Unwalled Maze 10x10

| Overall | | First Solve | | Steps to Solve | | |
|---|---|---|---|---|---|---|
| Agent | Solve Rate | Run | Memory size | Avg | Min | Max |
| Random | 8/100 | 8 | - | 76.8 | 54.0 | 100.0 |
| TD-Learning (Train) | 8/100 | 21 | 90 | 72.5 | 44.0 | 100.0 |
| TD-Learning (Test) | **100/100** | **1** | 90 | **18.0** | **18.0** | **18.0** |
| Q-Learning (Train) | 8/100 | 21 | 292 | 72.5 | 44.0 | 100.0 |
| Q-Learning (Test) | **100/100** | **1** | 294 | **18.0** | **18.0** | **18.0** |
| Go-Explore | 0/100 | - | - | - | - | - |
| Go-Explore GDIR | 38/100 | 12 | 59 | 83.3 | 62.0 | 100.0 |
| Go-Explore-Count | **100/100** | **1** | 71 | 62.0 | 62.0 | 62.0 |
| Go-Explore-Count GDIR | **100/100** | **1** | **39** | 20.0 | 20.0 | 20.0 |
| Explore-Count | **100/100** | **1** | 71 | 22.4 | 22.0 | 62.0 |
| Explore-Count GDIR | **100/100** | **1** | **39** | 20.0 | 20.0 | 20.0 |

Table 9: Performance results for Unwalled Maze 20x20

| Overall | | First Solve | | Steps to Solve | | |
|---|---|---|---|---|---|---|
| Agent | Solve Rate | Run | Memory size | Avg | Min | Max |
| Random | 0/100 | - | - | - | - | - |
| TD-Learning (Train) | 3/100 | 7 | 261 | 336.7 | 264.0 | 400.0 |
| TD-Learning (Test) | **100/100** | **1** | 359 | **38.0** | **38.0** | **38.0** |
| Q-Learning (Train) | 3/100 | 7 | 769 | 336.7 | 264.0 | 400.0 |
| Q-Learning (Test) | 0/100 | - | - | - | - | - |
| Go-Explore | 0/100 | - | - | - | - | - |
| Go-Explore GDIR | 0/100 | - | - | - | - | - |
| Go-Explore-Count | 99/100 | 2 | 359 | 356.0 | 356.0 | 356.0 |
| Go-Explore-Count GDIR | **100/100** | **1** | **85** | 42.0 | 42.0 | 42.0 |
| Explore-Count | 99/100 | 2 | 359 | 78.3 | 76.0 | 304.0 |
| Explore-Count GDIR | **100/100** | **1** | **85** | 42.0 | 42.0 | 42.0 |

Table 10: Performance results for Unwalled Maze 100x100

| Overall | | First Solve | | Steps to Solve | | |
|---|---|---|---|---|---|---|
| Agent | Solve Rate | Run | Memory size | Avg | Min | Max |
| Random | 1/100 | 15 | - | 9980.0 | 9980.0 | 9980.0 |
| TD-Learning (Train) | 2/100 | 15 | 8552 | 7606.0 | 5544.0 | 9668.0 |
| TD-Learning (Test) | **100/100** | **1** | 8997 | **198.0** | **198.0** | **198.0** |
| Q-Learning (Train) | 2/100 | 15 | 29289 | 7606.0 | 5544.0 | 9668.0 |
| Q-Learning (Test) | 0/100 | - | - | - | - | - |
| Go-Explore | 0/100 | - | - | - | - | - |
| Go-Explore GDIR | 0/100 | - | - | - | - | - |
| Go-Explore-Count | **100/100** | **1** | 8045 | 7810.0 | 7810.0 | 7810.0 |
| Go-Explore-Count GDIR | **100/100** | **1** | **499** | 230.0 | 230.0 | 230.0 |
| Explore-Count | **100/100** | **1** | 8045 | 555.3 | 482.0 | 7810.0 |
| Explore-Count GDIR | **100/100** | **1** | **499** | 222.1 | 222.0 | 230.0 |

Tables 11, 12 and 13 detail the results for various sizes of unwalled maze without hippocampal replay. A '-' represents unsolved after 100 runs, and bolded text means best result across all agents. It can be seen that hippocampal replay does not affect first solve run and memory size, since hippocampal replay only happens after a successful first solve. After the first solve, having no hippocampal replay causes the agent to explore instead of consolidating good experiences, which generally leads to lower solve rate and higher maximum solve time step. Increased exploration may be a good thing in the long run, as the agent generally finds a shorter solution than the corresponding agents with hippocampal replay. Here, some of the agents without hippocampal replay even manage to find the shortest path (18, 38, 198 for 10x10, 20x20 and 100x100 maze sizes respectively).

Table 11: Performance results for Unwalled Maze 10x10 (Without Hippocampal Replay)

| Overall | | First Solve | | Steps to Solve | | |
|---|---|---|---|---|---|---|
| Agent | Solve Rate | Run | Memory size | Avg | Min | Max |
| Go-Explore | 0/100 | - | - | - | - | - |
| Go-Explore GDIR | 5/100 | 12 | 59 | 82.8 | 64.0 | 100.0 |
| Go-Explore-Count | 82/100 | **1** | 71 | 50.9 | **18.0** | 100.0 |
| Go-Explore-Count GDIR | 78/100 | **1** | **39** | 42.0 | **18.0** | 100.0 |
| Explore-Count | **98/100** | **1** | 71 | **37.2** | **18.0** | **90.0** |
| Explore-Count GDIR | 92/100 | **1** | **39** | 38.2 | **18.0** | 98.0 |

Table 12: Performance results for Unwalled Maze 20x20 (Without Hippocampal Replay)

| Overall | | First Solve | | Steps to Solve | | |
|---|---|---|---|---|---|---|
| Agent | Solve Rate | Run | Memory size | Avg | Min | Max |
| Go-Explore | 0/100 | - | - | - | - | - |
| Go-Explore GDIR | 0/100 | - | - | - | - | - |
| Go-Explore-Count | 71/100 | 2 | 359 | 200.9 | **38.0** | 396.0 |
| Go-Explore-Count GDIR | 50/100 | **1** | **85** | 225.5 | 40.0 | **370.0** |
| Explore-Count | **79/100** | 2 | 359 | **178.1** | **38.0** | 400.0 |
| Explore-Count GDIR | **79/100** | **1** | **85** | 204.7 | **38.0** | 400.0 |

Table 13: Performance results for Unwalled Maze 100x100 (Without Hippocampal Replay)

| Overall | | First Solve | | Steps to Solve | | |
|---|---|---|---|---|---|---|
| Agent | Solve Rate | Run | Memory size | Avg | Min | Max |
| Go-Explore | 0/100 | - | - | - | - | - |
| Go-Explore GDIR | 0/100 | - | - | - | - | - |
| Go-Explore-Count | 94/100 | **1** | 8045 | 6367.6 | 5576.0 | 9980.0 |
| Go-Explore-Count GDIR | 32/100 | **1** | **499** | 7262.6 | 230.0 | 9976.0 |
| Explore-Count | **100/100** | **1** | 8045 | **2061.6** | 322.0 | **8028.0** |
| Explore-Count GDIR | 81/100 | **1** | **499** | 3640.2 | **198.0** | 9732.0 |

# F FULL RESULTS FOR WALLED MAZE

## F.1 WITH HIPPOCAMPAL REPLAY

Tables 14, 15 and 16 detail the results for various sizes of walled maze with hippocampal replay. A '-' represents unsolved after 100 runs, and bolded text means best result across all agents. Go-Explore-Count and Explore-Count and their GDIR variants perform the best.

Table 14: Performance results for Walled Maze 10x10

| Overall | | First Solve | | Steps to Solve | | |
|---|---|---|---|---|---|---|
| Agent | Solve Rate | Run | Memory size | Avg | Min | Max |
| Random | 0/100 | - | - | - | - | - |
| TD-Learning (Train) | 0/100 | - | - | - | - | - |
| TD-Learning (Test) | 0/100 | - | - | - | - | - |
| Q-Learning (Train) | 0/100 | - | - | - | - | - |
| Q-Learning (Test) | 0/100 | - | - | - | - | - |
| Go-Explore | 0/100 | - | - | - | - | - |
| Go-Explore GDIR | 4/100 | 75 | 77 | 94.0 | 92.0 | 100.0 |
| Go-Explore-Count | **100/100** | **1** | 74 | 64.0 | 64.0 | 64.0 |
| Go-Explore-Count GDIR | **100/100** | **1** | **62** | 60.0 | 60.0 | **60.0** |
| Explore-Count | **100/100** | **1** | 74 | 42.2 | 42.0 | 64.0 |
| Explore-Count GDIR | **100/100** | **1** | **62** | **34.3** | **34.0** | **60.0** |

Table 15: Performance results for Walled Maze 20x20

| Overall | | First Solve | | Steps to Solve | | |
|---|---|---|---|---|---|---|
| Agent | Solve Rate | Run | Memory size | Avg | Min | Max |
| Random | 0/100 | - | - | - | - | - |
| TD-Learning (Train) | 0/100 | - | - | - | - | - |
| TD-Learning (Test) | 0/100 | - | - | - | - | - |
| Q-Learning (Train) | 0/100 | - | - | - | - | - |
| Q-Learning (Test) | 0/100 | - | - | - | - | - |
| Go-Explore | 0/100 | - | - | - | - | - |
| Go-Explore GDIR | 0/100 | - | - | - | - | - |
| Go-Explore-Count | **100/100** | **1** | 312 | 218.0 | 218.0 | **218.0** |
| Go-Explore-Count GDIR | **100/100** | **1** | **223** | 220.0 | 220.0 | 220.0 |
| Explore-Count | **100/100** | **1** | 312 | 138.8 | 138.0 | **218.0** |
| Explore-Count GDIR | **100/100** | **1** | **223** | **83.4** | **82.0** | 220.0 |

Table 16: Performance results for Walled Maze 100x100

| Overall | | First Solve | | Steps to Solve | | |
|---|---|---|---|---|---|---|
| Agent | Solve Rate | Run | Memory size | Avg | Min | Max |
| Random | 0/100 | - | - | - | - | - |
| TD-Learning (Train) | 0/100 | - | - | - | - | - |
| TD-Learning (Test) | 0/100 | - | - | - | - | - |
| Q-Learning (Train) | 0/100 | - | - | - | - | - |
| Q-Learning (Test) | 0/100 | - | - | - | - | - |
| Go-Explore | 0/100 | - | - | - | - | - |
| Go-Explore GDIR | 0/100 | - | - | - | - | - |
| Go-Explore-Count | **100/100** | **1** | 7552 | 4912.0 | 4912.0 | 4912.0 |
| Go-Explore-Count GDIR | **100/100** | **1** | **5069** | 4042.0 | 4042.0 | **4042.0** |
| Explore-Count | **100/100** | **1** | 7552 | 3177.5 | 3160.0 | 4912.0 |
| Explore-Count GDIR | **100/100** | **1** | **5069** | **1450.2** | **1424.0** | **4042.0** |

## F.2 WITHOUT HIPPOCAMPAL REPLAY

Tables 11, 12 and 13 detail the results for various sizes of walled maze without hippocampal replay in the Go-Explore variants. A '-' represents unsolved after 100 runs, and bolded text means best result across all agents. It can be seen that hippocampal replay does not affect first solve run and memory size, since hippocampal replay only happens after a successful first solve. After the first solve, having no hippocampal replay causes the agent to explore instead of consolidating good experiences, which generally leads to lower solve rate and higher maximum solve time step. Increased exploration may be a good thing in the long run, as the agent generally finds a shorter solution than the corresponding agents with hippocampal replay.

Table 17: Performance results for Walled Maze 10x10 (Without Hippocampal Replay)

| Overall | | First Solve | | Steps to Solve | | |
|---|---|---|---|---|---|---|
| Agent | Solve Rate | Run | Memory size | Avg | Min | Max |
| Go-Explore | 0/100 | - | - | - | - | - |
| Go-Explore GDIR | 2/100 | 75 | 77 | 96.0 | 92.0 | **100.0** |
| Go-Explore-Count | **82/100** | **1** | 74 | 61.6 | **32.0** | **100.0** |
| Go-Explore-Count GDIR | 75/100 | **1** | **62** | **55.7** | **32.0** | **100.0** |
| Explore-Count | 75/100 | **1** | 74 | 61.6 | 34.0 | **100.0** |
| Explore-Count GDIR | 73/100 | **1** | **62** | 63.9 | 34.0 | **100.0** |

Table 18: Performance results for Walled Maze 20x20 (Without Hippocampal Replay)

| Overall | | First Solve | | Steps to Solve | | |
|---|---|---|---|---|---|---|
| Agent | Solve Rate | Run | Memory size | Avg | Min | Max |
| Go-Explore | 0/100 | - | - | - | - | - |
| Go-Explore GDIR | 0/100 | - | - | - | - | - |
| Go-Explore-Count | **87/100** | **1** | 312 | 233.1 | **68.0** | 400.0 |
| Go-Explore-Count GDIR | 77/100 | **1** | **223** | **221.6** | **68.0** | **394.0** |
| Explore-Count | 74/100 | **1** | 312 | 290.2 | 98.0 | 396.0 |
| Explore-Count GDIR | 48/100 | **1** | **223** | 299.5 | 74.0 | 396.0 |

Table 19: Performance results for Walled Maze 100x100 (Without Hippocampal Replay)

| Overall | | First Solve | | Steps to Solve | | |
|---|---|---|---|---|---|---|
| Agent | Solve Rate | Run | Memory size | Avg | Min | Max |
| Go-Explore | 0/100 | - | - | - | - | - |
| Go-Explore GDIR | 0/100 | - | - | - | - | - |
| Go-Explore-Count | **100/100** | **1** | 7552 | 4918.2 | 4718.0 | 6362.0 |
| Go-Explore-Count GDIR | **100/100** | **1** | **5069** | **3891.4** | 3676.0 | **5326.0** |
| Explore-Count | 32/100 | **1** | 7552 | 7376.2 | 3674.0 | 9954.0 |
| Explore-Count GDIR | 34/100 | **1** | **5069** | 4611.6 | **1410.0** | 9818.0 |

Tables 20, 21, 22, 23 and 24 detail the performance results of Explore-Count GDIR across 10 trials, with different ranges of random intrinsic rewards. A '-' represents unsolved after 100 runs. It can be seen that as the magnitude of uninformed (random) intrinsic reward increases, the overall solve rate decreases, first solve run increases, first solve memory size increases indicating more exploration of the environment. Explore-Count GDIR with random intrinsic rewards generally achieve a lower minimum solve time step compared to the actual Manhattan intrinsic reward for Walled Maze 100x100 (see Table 16), due to their increased exploration leading to a higher chance of finding a shorter path. This, however, comes at the cost of slower adaptation and sometimes even failing to solve the environment.

Table 20: Performance results for Explore-Count GDIR on Walled Maze 100x100 across 10 trials (Random Intrinsic Reward between $[-1, 0]$)

| Overall | | First Solve | | Steps to Solve | | |
|---|---|---|---|---|---|---|
| Trial | Solve Rate | Run | Memory size | Avg | Min | Max |
| 1 | 100/100 | 1 | 6771 | 497.9 | 414.0 | 8802.0 |
| 2 | 100/100 | 1 | 6541 | 537.1 | 472.0 | 6986.0 |
| 3 | 99/100 | 2 | 9236 | 849.1 | 798.0 | 5860.0 |
| 4 | 98/100 | 3 | 8871 | 702.3 | 640.0 | 6748.0 |
| 5 | 100/100 | 1 | 8457 | 598.0 | 504.0 | 9908.0 |
| 6 | 100/100 | 1 | 7001 | 558.3 | 474.0 | 8908.0 |
| 7 | 99/100 | 2 | 7592 | 1063.8 | 1004.0 | 6928.0 |
| 8 | 98/100 | 3 | 9607 | 1130.9 | 1082.0 | 5874.0 |
| 9 | 99/100 | 2 | 8132 | 706.3 | 652.0 | 6030.0 |
| 10 | 100/100 | 1 | 5048 | 560.1 | 510.0 | 5516.0 |
| **Avg** | **99.3/100** | **1.7** | **7725.6** | **720.4** | **655.0** | **7156.0** |

Table 21: Performance results for Explore-Count GDIR on Walled Maze 100x100 across 10 trials (Random Intrinsic Reward between $[-5, 0]$)

| Overall | | First Solve | | Steps to Solve | | |
|---|---|---|---|---|---|---|
| Trial | Solve Rate | Run | Memory size | Avg | Min | Max |
| 1 | 99/100 | 2 | 8643 | 881.8 | 838.0 | 5176.0 |
| 2 | 97/100 | 4 | 9653 | 587.9 | 492.0 | 9794.0 |
| 3 | 100/100 | 1 | 6778 | 531.9 | 460.0 | 7652.0 |
| 4 | 100/100 | 1 | 6603 | 527.8 | 466.0 | 6646.0 |
| 5 | 99/100 | 2 | 9178 | 591.6 | 504.0 | 9180.0 |
| 6 | 100/100 | 1 | 5428 | 487.8 | 426.0 | 6602.0 |
| 7 | 99/100 | 2 | 9181 | 753.0 | 662.0 | 9674.0 |
| 8 | 100/100 | 1 | 6865 | 503.0 | 434.0 | 7336.0 |
| 9 | 98/100 | 3 | 8970 | 865.6 | 790.0 | 8198.0 |
| 10 | 99/100 | 2 | 8618 | 588.8 | 500.0 | 9288.0 |
| **Avg** | **99.1/100** | **1.9** | **7991.7** | **631.9** | **557.2** | **7954.6** |

Table 22: Performance results for Explore-Count GDIR on Walled Maze 100x100 across 10 trials (Random Intrinsic Reward between $[-10, 0]$)

| | Overall | First Solve | | Steps to Solve | | |
|---|---|---|---|---|---|---|
| **Trial** | **Solve Rate** | **Run** | **Memory size** | **Avg** | **Min** | **Max** |
| 1 | 100/100 | 1 | 5353 | 551.8 | 506.0 | 5082.0 |
| 2 | 99/100 | 2 | 8042 | 537.3 | 468.0 | 7328.0 |
| 3 | 99/100 | 2 | 7976 | 1022.1 | 990.0 | 4164.0 |
| 4 | 98/100 | 3 | 9562 | 1513.7 | 1486.0 | 4198.0 |
| 5 | 100/100 | 1 | 7368 | 573.0 | 494.0 | 8394.0 |
| 6 | 98/100 | 3 | 8668 | 1758.1 | 1754.0 | 2152.0 |
| 7 | 100/100 | 1 | 7610 | 499.7 | 416.0 | 8784.0 |
| 8 | 100/100 | 1 | 6220 | 535.6 | 478.0 | 6238.0 |
| 9 | 100/100 | 1 | 6889 | 511.3 | 416.0 | 9942.0 |
| 10 | 99/100 | 2 | 8610 | 638.9 | 612.0 | 3274.0 |
| **Avg** | **99.3/100** | **1.7** | **7629.8** | **814.2** | **762.0** | **5955.6** |

Table 23: Performance results for Explore-Count GDIR on Walled Maze 100x100 across 10 trials (Random Intrinsic Reward between $[-20, 0]$)

| | Overall | First Solve | | Steps to Solve | | |
|---|---|---|---|---|---|---|
| **Trial** | **Solve Rate** | **Run** | **Memory size** | **Avg** | **Min** | **Max** |
| 1 | 85/100 | 16 | 9707 | 1293.3 | 1230.0 | 5034.0 |
| 2 | 87/100 | 14 | 9687 | 774.1 | 660.0 | 9046.0 |
| 3 | 95/100 | 6 | 9403 | 707.7 | 634.0 | 7632.0 |
| 4 | 95/100 | 6 | 9378 | 744.8 | 662.0 | 8506.0 |
| 5 | 96/100 | 5 | 9668 | 528.7 | 454.0 | 7624.0 |
| 6 | 91/100 | 10 | 9706 | 749.5 | 650.0 | 9702.0 |
| 7 | 89/100 | 12 | 9707 | 672.3 | 574.0 | 9320.0 |
| 8 | 96/100 | 5 | 9621 | 781.6 | 712.0 | 7392.0 |
| 9 | 81/100 | 20 | 9705 | 1244.0 | 1158.0 | 8124.0 |
| 10 | 90/100 | 11 | 9707 | 667.1 | 570.0 | 9308.0 |
| **Avg** | **90.5/100** | **10.5** | **9628.9** | **816.3** | **730.4** | **8168.8** |

Table 24: Performance results for Explore-Count GDIR on Walled Maze 100x100 across 10 trials (Random Intrinsic Reward between $[-50, 0]$, '-' means unsolved). If unsolved after 100 runs, we will count it as the first solve occuring at the $100^{th}$ run for the average score.

| | Overall | First Solve | | Steps to Solve | | |
|---|---|---|---|---|---|---|
| **Trial** | **Solve Rate** | **Run** | **Memory size** | **Avg** | **Min** | **Max** |
| 1 | 0/100 | - | - | - | - | - |
| 2 | 0/100 | - | - | - | - | - |
| 3 | 0/100 | - | - | - | - | - |
| 4 | 14/100 | 87 | 9707 | 1082.0 | 532.0 | 7972.0 |
| 5 | 0/100 | - | - | - | - | - |
| 6 | 0/100 | - | - | - | - | - |
| 7 | 0/100 | - | - | - | - | - |
| 8 | 0/100 | - | - | - | - | - |
| 9 | 26/100 | 75 | 9678 | 941.8 | 590.0 | 9418.0 |
| 10 | 0/100 | - | - | - | - | - |
| **Avg** | **4/100** | **96.2** | **9692.5** | **1011.9** | **561** | **8695** |

## G  FULL RESULTS FOR TOWERS OF HANOI

Tables 25 and 26 detail the results for Towers of Hanoi with 3 and 7 disks respectively. A '-' represents unsolved after 100 runs, and bolded text means best result across all agents. From the results, we can see that Explore-Count and Explore-Count GDIR perform the best and are the only agents which can solve the task with 7 disks.

Table 25: Performance results for Towers of Hanoi with 3 disks

| Overall | | First Solve | | Steps to Solve | | |
|---|---|---|---|---|---|---|
| Agent | Solve Rate | Run | Memory size | Avg | Min | Max |
| Random | 10/100 | 8 | - | 25.4 | 17.0 | 32.0 |
| TD-Learning (Train) | 13/100 | 4 | 21 | 24.9 | 13.0 | 32.0 |
| TD-Learning (Test) | **100/100** | **1** | 27 | **7.0** | **7.0** | **7.0** |
| Q-Learning (Train) | 13/100 | 4 | 46 | 24.9 | 13.0 | 32.0 |
| Q-Learning (Test) | **100/100** | **1** | 78 | **7.0** | **7.0** | **7.0** |
| Go-Explore | 41/100 | 33 | 27 | 20.2 | 13.0 | 31.0 |
| Go-Explore GDIR | 41/100 | 33 | 27 | 20.2 | 13.0 | 31.0 |
| Go-Explore-Count | **100/100** | **1** | **15** | 9.0 | 9.0 | 9.0 |
| Go-Explore-Count GDIR | **100/100** | **1** | **15** | 9.0 | 9.0 | 9.0 |
| Explore-Count | **100/100** | **1** | **15** | 9.0 | 9.0 | 9.0 |
| Explore-Count GDIR | **100/100** | **1** | **15** | 9.0 | 9.0 | 9.0 |

Table 26: Performance results for Towers of Hanoi with 7 disks

| Overall | | First Solve | | Steps to Solve | | |
|---|---|---|---|---|---|---|
| Agent | Solve Rate | Run | Memory size | Avg | Min | Max |
| Random | 0/100 | - | - | - | - | - |
| TD-Learning (Train) | 0/100 | - | - | - | - | - |
| TD-Learning (Test) | 0/100 | - | - | - | - | - |
| Q-Learning (Train) | 0/100 | - | - | - | - | - |
| Q-Learning (Test) | 0/100 | - | - | - | - | - |
| Go-Explore | 0/100 | - | - | - | - | - |
| Go-Explore GDIR | 0/100 | - | - | - | - | - |
| Go-Explore-Count | 0/100 | - | - | - | - | - |
| Go-Explore-Count GDIR | 0/100 | - | - | - | - | - |
| Explore-Count | 84/100 | 17 | 1720 | 378.6 | 378.0 | 426.0 |
| Explore-Count GDIR | **91/100** | **10** | **1095** | **327.1** | **327.0** | **335.0** |

## H  FULL RESULTS FOR NIM

Tables 27, 28 and 29 detail the results for Nim with 11, 21 and 1001 matchsticks respectively. A '-' represents unsolved after 100 runs, and bolded text means best result across all agents. From the results, we see that Go-Explore-Count and Explore-Count, as well as their GDIR variants perform the best and are able to solve the complex Nim environment with 1001 matchsticks.

Table 27: Performance results for Nim with 11 matches

| Overall | | First Solve | | Steps to Solve | | |
|---|---|---|---|---|---|---|
| Agent | Solve Rate | Run | Memory size | Avg | Min | Max |
| Random | 4/100 | 71 | - | **3.0** | **3.0** | **3.0** |
| TD-Learning (Train) | 5/100 | 27 | **7** | **3.0** | **3.0** | **3.0** |
| TD-Learning (Test) | **100/100** | **1** | **7** | **3.0** | **3.0** | **3.0** |
| Q-Learning (Train) | 5/100 | 27 | 17 | **3.0** | **3.0** | **3.0** |
| Q-Learning (Test) | **100/100** | **1** | 17 | **3.0** | **3.0** | **3.0** |
| Go-Explore | 15/100 | 7 | **7** | **3.0** | **3.0** | **3.0** |
| Go-Explore GDIR | 16/100 | 6 | **7** | **3.0** | **3.0** | **3.0** |
| Go-Explore-Count | 99/100 | 2 | **7** | **3.0** | **3.0** | **3.0** |
| Go-Explore-Count GDIR | **100/100** | **1** | **7** | **3.0** | **3.0** | **3.0** |
| Explore-Count | 99/100 | 2 | **7** | **3.0** | **3.0** | **3.0** |
| Explore-Count GDIR | **100/100** | **1** | **7** | **3.0** | **3.0** | **3.0** |

Table 28: Performance results for Nim with 21 matches

| Overall | | First Solve | | Steps to Solve | | |
|---|---|---|---|---|---|---|
| Agent | Solve Rate | Run | Memory size | Avg | Min | Max |
| Random | 0/100 | - | - | - | - | - |
| TD-Learning (Train) | 0/100 | - | - | - | - | - |
| TD-Learning (Test) | 0/100 | - | - | - | - | - |
| Q-Learning (Train) | 0/100 | - | - | - | - | - |
| Q-Learning (Test) | 0/100 | - | - | - | - | - |
| Go-Explore | 16/100 | 5 | **12** | **6.0** | **6.0** | **6.0** |
| Go-Explore GDIR | 16/100 | 5 | **12** | **6.0** | **6.0** | **6.0** |
| Go-Explore-Count | **99/100** | **2** | **12** | **6.0** | **6.0** | **6.0** |
| Go-Explore-Count GDIR | **99/100** | **2** | **12** | **6.0** | **6.0** | **6.0** |
| Explore-Count | 98/100 | 3 | **12** | **6.0** | **6.0** | **6.0** |
| Explore-Count GDIR | **99/100** | **2** | **12** | **6.0** | **6.0** | **6.0** |

Table 29: Performance results for Nim with 1001 matches

| Overall | | First Solve | | Steps to Solve | | |
|---|---|---|---|---|---|---|
| Agent | Solve Rate | Run | Memory size | Avg | Min | Max |
| Random | 0/100 | - | - | - | - | - |
| TD-Learning (Train) | 0/100 | - | - | - | - | - |
| TD-Learning (Test) | 0/100 | - | - | - | - | - |
| Q-Learning (Train) | 0/100 | - | - | - | - | - |
| Q-Learning (Test) | 0/100 | - | - | - | - | - |
| Go-Explore | 0/100 | - | - | - | - | - |
| Go-Explore GDIR | 0/100 | - | - | - | - | - |
| Go-Explore-Count | **99/100** | **2** | 502 | **251.0** | **251.0** | **251.0** |
| Go-Explore-Count GDIR | **99/100** | **2** | 502 | **251.0** | **251.0** | **251.0** |
| Explore-Count | 98/100 | 3 | 502 | **251.0** | **251.0** | **251.0** |
| Explore-Count GDIR | **99/100** | **2** | 502 | **251.0** | **251.0** | **251.0** |

# I FULL RESULTS FOR CART POLE

Tables 30, 32 and 34 detail the results for Cart Pole for 50, 100 and 175 goal timesteps respectively. Tables 31, 33 and 35 detail the results for Cart Pole with distinct repeated states for 50, 100 and 175 goal timesteps respectively. Table 36 detail the results for PPO for various goal timesteps of Cart Pole with discretized (same representation as the other agents) and continuous state space inputs. A '-' represents unsolved after 100 runs, and bolded text means best result across all agents.

**Overall.** From the results, we can see that as the Cart Pole goal timesteps to balance increases, the number of algorithms which can solve it decreases. Overall, the first solve run is increased as the difficulty increases, which is intuitive as it is harder to find a successful trajectory as the trajectory becomes longer. GDIR variants tend to solve the environment faster in general, except for Explore-Count which tends to make it worse, which we suspect is due to GDIR causing Explore-Count to repeat the same states again where the pole is balanced, hence leading to a high visit count. This leads it to avoid these states in the future, which make it take longer to solve.

**Repeated States.** From the results, we can see that having repeated states lead to generally increased solve rate for the Go-Explore variants, and lower first solve run. While it helped to boost performance in Cart Pole, having repeated states will lead to increasing the search space, which can lead to larger first solve memory size and may not be viable for large state spaces.

**PPO.** Overall, PPO performs well even in the sparse reward modification environment of Cart Pole. Similar to the other algorithms, PPO also has a higher first solve run the higher the goal timesteps to balance for Cart Pole. Moreover, having a discretized state representation helps PPO get a lower first solve run, which highlights that an appropriately discretized state space can enable the neural network to learn faster.

Table 30: Performance results for Cart Pole with 50 timesteps (binary reward)

| Overall | | First Solve | | Steps to Solve | | |
|---|---|---|---|---|---|---|
| **Agent** | **Solve Rate** | **Run** | **Memory size** | **Avg** | **Min** | **Max** |
| Random | 2/100 | 57 | - | 53.0 | 51.0 | 55.0 |
| TD-Learning (Train) | 2/100 | 8 | 173 | 54.5 | 52.0 | 57.0 |
| TD-Learning (Test) | 0/100 | - | - | - | - | - |
| Q-Learning (Train) | 2/100 | 8 | 272 | 54.5 | 52.0 | 57.0 |
| Q-Learning (Test) | 0/100 | - | - | - | - | - |
| Go-Explore | 97/100 | 4 | 91 | 51.0 | 51.0 | 51.0 |
| Go-Explore GDIR | 97/100 | 4 | 110 | 84.0 | 84.0 | 84.0 |
| Go-Explore-Count | **98/100** | **3** | 59 | 61.0 | 61.0 | 61.0 |
| Go-Explore-Count GDIR | **98/100** | **3** | **54** | 53.0 | 53.0 | 53.0 |
| Explore-Count | 82/100 | **3** | 78 | 57.1 | 52.0 | 61.0 |
| Explore-Count GDIR | 77/100 | 4 | 89 | 62.2 | 51.0 | 63.0 |

Table 31: Performance results for Cart Pole with 50 timesteps and repeated states (binary reward)

| Overall | | First Solve | | Steps to Solve | | |
|---|---|---|---|---|---|---|
| Agent | Solve Rate | Run | Memory size | Avg | Min | Max |
| Random | 2/100 | 57 | - | 53.0 | 51.0 | 55.0 |
| TD-Learning (Train) | 2/100 | 8 | 196 | 54.5 | 52.0 | 57.0 |
| TD-Learning (Test) | 0/100 | - | - | - | - | - |
| Q-Learning (Train) | 2/100 | 8 | 306 | 54.5 | 52.0 | 57.0 |
| Q-Learning (Test) | 0/100 | - | - | - | - | - |
| Go-Explore | 97/100 | 4 | 98 | 51.0 | 51.0 | 51.0 |
| Go-Explore GDIR | 97/100 | 4 | 127 | 92.2 | 75.0 | 93.0 |
| Go-Explore-Count | **99/100** | **2** | **86** | 76.0 | 76.0 | 76.0 |
| Go-Explore-Count GDIR | **99/100** | **2** | **86** | 129.3 | 76.0 | 130.0 |
| Explore-Count | 49/100 | 4 | 204 | 159.0 | 159.0 | 159.0 |
| Explore-Count GDIR | 53/100 | 4 | 204 | 102.1 | 96.0 | 159.0 |

Table 32: Performance results for Cart Pole with 100 timesteps (binary reward)

| Overall | | First Solve | | Steps to Solve | | |
|---|---|---|---|---|---|---|
| Agent | Solve Rate | Run | Memory size | Avg | Min | Max |
| Random | 0/100 | - | - | - | - | - |
| TD-Learning (Train) | 0/100 | - | - | - | - | - |
| TD-Learning (Test) | 0/100 | - | - | - | - | - |
| Q-Learning (Train) | 0/100 | - | - | - | - | - |
| Q-Learning (Test) | 0/100 | - | - | - | - | - |
| Go-Explore | 79/100 | 22 | 199 | 113.0 | 113.0 | 113.0 |
| Go-Explore GDIR | **95/100** | **6** | 204 | 114.0 | 114.0 | 114.0 |
| Go-Explore-Count | 74/100 | 27 | 185 | 101.0 | 101.0 | 101.0 |
| Go-Explore-Count GDIR | **95/100** | **6** | **111** | 104.0 | 104.0 | 104.0 |
| Explore-Count | 2/100 | 56 | 725 | 109.5 | 108.0 | 111.0 |
| Explore-Count GDIR | 1/100 | 69 | 897 | 107.0 | 107.0 | 107.0 |

Table 33: Performance results for Cart Pole with 100 timesteps and repeated states (binary reward)

| Overall | | First Solve | | Steps to Solve | | |
|---|---|---|---|---|---|---|
| Agent | Solve Rate | Run | Memory size | Avg | Min | Max |
| Random | 0/100 | - | - | - | - | - |
| TD-Learning (Train) | 0/100 | - | - | - | - | - |
| TD-Learning (Test) | 0/100 | - | - | - | - | - |
| Q-Learning (Train) | 0/100 | - | - | - | - | - |
| Q-Learning (Test) | 0/100 | - | - | - | - | - |
| Go-Explore | 79/100 | 22 | 206 | 106.0 | 106.0 | 106.0 |
| Go-Explore GDIR | 95/100 | 6 | 257 | 178.4 | 110.0 | 188.0 |
| Go-Explore-Count | **96/100** | 5 | **140** | 103.0 | 103.0 | 103.0 |
| Go-Explore-Count GDIR | **96/100** | 5 | 155 | 113.0 | 113.0 | 113.0 |
| Explore-Count | 49/100 | **4** | 204 | 159.0 | 159.0 | 159.0 |
| Explore-Count GDIR | 58/100 | **4** | 204 | 141.4 | 130.0 | 159.0 |

Table 34: Performance results for Cart Pole with 175 timesteps (binary reward)

| Overall | | First Solve | | Steps to Solve | | |
|---|---|---|---|---|---|---|
| **Agent** | **Solve Rate** | **Run** | **Memory size** | **Avg** | **Min** | **Max** |
| Random | 0/100 | - | - | - | - | - |
| TD-Learning (Train) | 0/100 | - | - | - | - | - |
| TD-Learning (Test) | 0/100 | - | - | - | - | - |
| Q-Learning (Train) | 0/100 | - | - | - | - | - |
| Q-Learning (Test) | 0/100 | - | - | - | - | - |
| Go-Explore | 0/100 | - | - | - | - | - |
| Go-Explore GDIR | **81/100** | **20** | 526 | 187.0 | 187.0 | 187.0 |
| Go-Explore-Count | 0/100 | - | - | - | - | - |
| Go-Explore-Count GDIR | 73/100 | 28 | **446** | 177.0 | 177.0 | 177.0 |
| Explore-Count | 0/100 | - | - | - | - | - |
| Explore-Count GDIR | 0/100 | - | - | - | - | - |

Table 35: Performance results for Cart Pole with 175 timesteps and repeated states (binary reward)

| Overall | | First Solve | | Steps to Solve | | |
|---|---|---|---|---|---|---|
| **Agent** | **Solve Rate** | **Run** | **Memory size** | **Avg** | **Min** | **Max** |
| Random | 0/100 | - | - | - | - | - |
| TD-Learning (Train) | 0/100 | - | - | - | - | - |
| TD-Learning (Test) | 0/100 | - | - | - | - | - |
| Q-Learning (Train) | 0/100 | - | - | - | - | - |
| Q-Learning (Test) | 0/100 | - | - | - | - | - |
| Go-Explore | 50/100 | 51 | **360** | 178.1 | 178.0 | 180.0 |
| Go-Explore GDIR | **76/100** | **21** | 620 | 198.5 | 178.0 | 200.0 |
| Go-Explore-Count | 75/100 | 26 | 412 | 194.0 | 194.0 | 194.0 |
| Go-Explore-Count GDIR | 54/100 | 40 | 545 | 180.3 | 176.0 | 190.0 |
| Explore-Count | 36/100 | 30 | 471 | 200.0 | 200.0 | 200.0 |
| Explore-Count GDIR | 7/100 | 75 | 1089 | 200.0 | 200.0 | 200.0 |

Table 36: First Solve results across 10 sets of 100 runs for PPO on Cart Pole. If unsolved after 100 runs, we will count it as the first solve occuring at the $100^{th}$ run for the average score.

| | Set | | | | | | | | | | |
|---|---|---|---|---|---|---|---|---|---|---|---|
| **Environment** | **1** | **2** | **3** | **4** | **5** | **6** | **7** | **8** | **9** | **10** | **Average** |
| 50 (Discrete) | 1 | 8 | 2 | 1 | 6 | 4 | 5 | 3 | 4 | 3 | **3.7** |
| 100 (Discrete) | 1 | 99 | 16 | 1 | 3 | 7 | 34 | 94 | - | 42 | **39.7** |
| 175 (Discrete) | 1 | - | 63 | 1 | 30 | - | 28 | - | - | - | **62.3** |
| 50 (Continuous) | 9 | 11 | 4 | 4 | 4 | 9 | 8 | 5 | 5 | 7 | **6.6** |
| 100 (Continuous) | 14 | - | 12 | 5 | - | 16 | - | 6 | - | - | **55.3** |
| 175 (Continuous) | - | - | 23 | 8 | - | 14 | 47 | 8 | - | - | **60.0** |

## J  FULL RESULTS FOR MOUNTAIN CAR

Tables 37 and 38 detail the results for Mountain Car with normal actions and with 10 repeated actions respectively. Table 39 detail the results for PPO for the normal and repeated actions version of Mountain Car with discretized (same representation as the other agents) and continuous state space inputs. A '-' represents unsolved after 100 runs, and bolded text means best result across all agents.

**Overall.** From the results, we can see that Explore-Count GDIR is the only algorithm which solves Mountain Car, indicating that it is indeed a difficult problem. If we were to relax the constraints and allow for 10 repeated actions (to reduce the search space), we can see that all Go-Explore variants with GDIR can solve the environment, with Go-Explore GDIR performing the best.

**PPO.** Overall, PPO performs well in the sparse reward modification environment of Mountain Car. Having a discretized state representation helps PPO get a lower first solve run, which highlights that an appropriately discretized state space can enable the neural network to learn faster.

Table 37: Performance results for Mountain Car (binary reward)

| Overall | | First Solve | | Steps to Solve | | |
|---|---|---|---|---|---|---|
| Agent | Solve Rate | Run | Memory size | Avg | Min | Max |
| Random | 0/100 | - | - | - | - | - |
| TD-Learning (Train) | 0/100 | - | - | - | - | - |
| TD-Learning (Test) | 0/100 | - | - | - | - | - |
| Q-Learning (Train) | 0/100 | - | - | - | - | - |
| Q-Learning (Test) | 0/100 | - | - | - | - | - |
| Go-Explore | 0/100 | - | - | - | - | - |
| Go-Explore GDIR | 0/100 | - | - | - | - | - |
| Go-Explore-Count | 0/100 | - | - | - | - | - |
| Go-Explore-Count GDIR | 0/100 | - | - | - | - | - |
| Explore-Count | 0/100 | - | - | - | - | - |
| Explore-Count GDIR | **7/100** | **3** | **862** | **165.4** | **156.0** | **187.0** |

Table 38: Performance results for Mountain Car with 10 repeated actions (binary reward)

| Overall | | First Solve | | Steps to Solve | | |
|---|---|---|---|---|---|---|
| Agent | Solve Rate | Run | Memory size | Avg | Min | Max |
| Random | 6/100 | 47 | - | 181.7 | 160.0 | 200.0 |
| TD-Learning (Train) | 5/100 | 17 | 231 | **166.0** | **120.0** | 200.0 |
| TD-Learning (Test) | 0/100 | - | - | - | - | - |
| Q-Learning (Train) | 5/100 | 17 | 324 | **166.0** | **120.0** | 200.0 |
| Q-Learning (Test) | 0/100 | - | - | - | - | - |
| Go-Explore | 63/100 | 38 | 89 | 170.0 | 170.0 | **170.0** |
| Go-Explore GDIR | **99/100** | **2** | **41** | 170.0 | 170.0 | **170.0** |
| Go-Explore-Count | 0/100 | - | - | - | - | - |
| Go-Explore-Count GDIR | 29/100 | 72 | 158 | 190.0 | 190.0 | 190.0 |
| Explore-Count | 91/100 | 10 | 165 | 180.0 | 180.0 | 180.0 |
| Explore-Count GDIR | 98/100 | 3 | 93 | 170.2 | 170.0 | 180.0 |

Table 39: First Solve results across 10 sets of 100 runs for PPO on Mountain Car. If unsolved after 100 runs, we will count it as the first solve occuring at the $100^{th}$ run for the average score.

| State | Set | | | | | | | | | | Average |
|---|---|---|---|---|---|---|---|---|---|---|---|
| | **1** | **2** | **3** | **4** | **5** | **6** | **7** | **8** | **9** | **10** | |
| Discrete (Normal) | - | 9 | - | 7 | - | 4 | 7 | - | 1 | 3 | **43.1** |
| Continuous (Normal) | - | - | - | - | 69 | - | - | - | - | 57 | **92.6** |
| Discrete (Repeated) | 2 | 1 | 6 | 2 | 5 | 1 | 4 | 5 | 3 | 1 | **3.0** |
| Continuous (Repeated) | 15 | 12 | 17 | 12 | 10 | 12 | 12 | 11 | 11 | 13 | **12.5** |

