# OpenReview forum: "Go-Explore with a guide: Speeding up search in sparse reward settings with goal-directed intrinsic rewards"
_ICLR.cc/2023/Conference — Submitted to ICLR 2023_

### Official Review · Reviewer_eHGy · 2022-10-23

**Confidence:** 5
**Correctness:** 1
**Technical Novelty And Significance:** 1
**Empirical Novelty And Significance:** 2
**Recommendation:** 1

**Clarity, Quality, Novelty And Reproducibility:**

## Clarity
The paper isn't very clear. See above regarding the motivation of the paper and its positioning in the related work. Most importantly, it's not clear what exact problem their approach is solving and how it compares with similar exploration methods.

## Quality
The paper lacks clear motivation and positioning in the broader literature, thorough experimental evaluation, and makes claims that are not well-supported by the experiments. The method also seems to have significant limitations that are not addressed, evaluated, or discussed in depth.

## Novelty
The approach isn't particularly novel, but merely a naive extension of Go-Explore and combination of potential-based reward shaping and goal-conditioned RL. However, I wouldn't use this as a reason to reject the paper if the other issues are addressed.

## Reproducibility
I don't see any mention of open-sourcing the code. I think it would be difficult to reproduce the paper without the code so I strongly recommend the authors to open-source it and in any case tell use their plans regarding this.

**Strength And Weaknesses:**

# Strengths:
- aims to solve an important problem, namely exploration in RL
- the method seems simple and intuitive

# Weaknesses:
### Claims are not well-supported
- In the introduction, you claim that your approach "learns faster and adapts better to novel environments in real-world system". However, most of these aren't well supported by your experiments. When you say it adapts better to novel environments, I imagine that you train it on some environments and finetune it on new ones, which you don't seem to do. Finally, you definitely don't show any results on real-world systems. In fact, it seems like your method only works in deterministic domains where you can reset the environment in any state, which isn't the case in the real-world. I suggest modifying the claim to be better supported by experiments.

- In the introduction, you claim that GDIR improves over Go-Explore. However, your experiments seem to suggest that it is merely better at first finding a successful trajectory rather than consistently maximizing the extrinsic reward and finding the optimal policy. Again, this claim needs to be more precise to be accurate.

- In the abstract, you discuss that Go-Explore is helpful in sparse-reward domains. However, all the environments you consider seem to have dense rewards. I recommend evaluating GDIR on environments such as Montezuma's Revenge, Pitfall, MiniHack, NetHack etc or at least demonstrating the sparsity of the reward in your settings.

### Approach is not well-motivated
- I don't understand the motivation for finding a satisficing solution rather than an optimal one. If your agent gets lucky once and finds a good path that achieves high extrinsic reward, there is no guarantee that it will robustly be able to solve the task (particularly in stochastic environments, see below) or that it is anywhere close to the optimal policy which may require much more exploration and exploitation to collect additional reward.

- The idea that you want an intrinsic reward that always encourages exploration in a setting where the goal is to maximize extrinsic reward is quite odd. If your intrinsic reward doesn't diminish throughout training, you will end up with a suboptimal policy wrt the extrinsic reward. Hence, the motivation for your approach doesn't make much sense in this setting. It might make more sense to apply GDIR in reward-free settings or open-ended learning with non-stationary extrinsic reward that requires constant exploration.

- The hippocampal replay trick is also quite limited given that it probably doesn't work well in stochastic environments. Due to this, it seems like the approach is limited to a fairly narrow class of problems which are not very challenging. In addition, it seems to require resetting the environment in any state which is a very strong assumption that doesn't apply in the real-world. I would like to see the approach evaluated in a wider range of domains, including stochastic ones.

- The introduction of this paper reads as if this is the first work to propose a type of intrinsic reward, cites LeCun 2022 for introducing ideas that have been well-known way before 2022 and have been introduced by others, cites neuroscience studies that are only remotely related to the current paper, instead of much more closely related works on intrinsic motivation and exploration in RL. Hence, it ignores a huge body of related work.

- GDIR isn't discussed in comparison with closely related works (other than Go-Explore) so it is unclear how it compares with HER, AMIGo, or other goal-conditioned exploration methods. What problem with existing methods is GDIR solving? This should be made more clear in the introduction.

- In general, there should be more discussion about the limitations of your approach, together with experiments on settings where you expect it to work best / worst.

### Empirical evaluation is limited
- GDIR is only compared with vanilla RL and a few ablations based on Go-Explore. The evaluations are not thorough and extensive enough to draw strong conclusions. For example, I would like to see a comparison with HER and other popular goal-based or potential-based methods. What about using reward shaping of the extrinsic reward?

- The authors include a model-based baseline but I'm not sure what this is meant to achieve since it's completely different from the model-free ones.

- It seems like GDIR makes use of inductive biases in its cost function e.g. Manhattan distance etc. This results in unfair comparisons. What happens if you also augment the other baselines with reward shaped based on such distances or include these representations in their states. If the agent knows where it is with respect to the goal, maybe it can more quickly find it even if it doesn't explore via GDIR explicitly.

- Important ablations seem to be missing. What happens if just apply goal-conditioned exploration without reward shaping or just reward shaping to the external reward without goal-conditioning?


**Summary Of The Paper:**

The paper proposes a potential-based goal-directed intrinsic reward to improve exploration and name it GDIR. By augmenting Go-Explore with GDIR, they improve its performance on a few discrete and continuous exploration methods. They also use a hippocampal replay to directly imitate successful trajectories.

**Summary Of The Review:**

This paper requires more work to warrant acceptance at the ICLR conference. In particular, the approach needs to be better positioned in the broader literature. The claims need to be toned down or better supported by the experiments. The approach could also use more work to alleviate clear limitations in more challenging environments or realistic settings, or at least these limitations should be openly discussed and analyzed. Finally, I am not sure the current experiments contain fair comparisons and could greatly benefit from more baselines, ablations, and environments.

---

> ### Author Response · Authors · 2022-11-18
> **Author Response**
>
> Thank you for the feedback. We would like to give our views on the comments mentioned in the review.
>
> Open-Sourcing the Code:
> We intend to open-source the code and will include this line in our revised paper.
>
> In the section “Claims are not well-supported”:
>
> 1.	The training on the environment is actually the fine-tuning process. The agents with GDIR can solve the environment (for instance Walled Maze 100x100) on the first-try, which is actually pretty amazing given that traditional approaches like TD-Learning and Q-Learning are unable to. Basically, no prior knowledge is needed before running through the environment and the agent can solve it quickly, which mimics how humans may solve a new environment fast without prior knowledge. Of course, incorporating prior knowledge in some form of memory transfer may make the method work faster on similar environments.
>
> 2.	It is indeed a limitation that the current method only works on deterministic domains. The start state and the goal state needs to be the same. It would be interesting to come up with a mechanism which can allow the re-use of the same memory for different start state and goal states. We believe that such an approach may require the use of a neural network (maybe with Transformers) to learn and retrieve memory, and intend to explore it in future work.
>
> 3.	We will tone down on the claims in the introduction and highlight that this method does not find seek to find the optimal policy.
>
> 4.	The environments considered are actually sparse in rewards. You only get a reward of 1 upon reaching the goal state, and a reward of 0 otherwise. This is extremely difficult for a RL agent to learn from, as there is no intermediate reward to guide it towards the goal state.
>
> In the section “Approach is not well-motivated”:
>
> 1.	The satisficing solution is actually inspired from real life. In real life, the agents do not have thousands of simulated time to adapt and learn from the environment, they actually only have one life and a very limited amount of time to gain experience. Hence, in contrast to majority of RL algorithms, we seek not to maximize the reward, but just to solve the environment and try to repeat this successful trajectory as much as possible using hippocampal replay. This will not lead to the most optimal solution, but at least the agent will be more survivable and can learn and remember successful solutions.
>
> 2.	The intrinsic reward actually does not encourage exploration – it is there to just guide initial action selection in a new state. The exploration term is actually numvisits and numselected in Equation (1).
>
> 3.	We intentionally did not choose to use stochastic rewards or stochastic environments, as these conditions force the agent to explore extensively in order to find the equilibrium state (e.g. mean reward, transition probabilities), which we feel current RL methods like Deep Q-Networks or Policy Gradients does adequately well. For scenarios whereby it is largely deterministic like in short-term real-life prediction or navigation problems, existing RL methods are highly inefficient as they need to explore extensively even after finding a solution and we seek to address the problems of this nature.
>
> In the section “Empirical evaluation is limited”:
>
> 1.	Using reward shaping of the extrinsic reward will be similar to GDIR, and we posit that the reason why this reward shaping works (e.g. in AlphaTensor) is because it is doing what this goal-directed intrinsic reward is doing in a new environment.
>
> 2.	The model-based one-step lookahead in the Q-Learning, TD-Learning and Go-Explore variants are just meant to speed up finding a solution in the new environment. It can be akin to doing planning of one time step ahead at every state. This can help direct the agent to go towards more promising states without the need to try out every state in real life (the agent tries it out in its head).
>
> 3.	We agree that the intrinsic rewards as of now are hand-crafted for each environment, but in the future, we intend to generalize this using some form of generic distance metric such as cosine similarity between states. This will be explored in future work.
>
> 4.	As mentioned in Part 2) of the previous paragraph, there is no goal-directed exploration here. The goal-directed part is merely to help choose an initial more promising state to go towards when the agent is not aware of its environment yet.
>
> Thank you once again for your review!

---

> > ### Comment · Reviewer_eHGy · 2022-11-24
> > **Post-Rebuttal Response**
> >
> > Thank you for your response.
> >
> > However, the paper hasn't been updated to change the claims to be more inline with what is shown by the experiments and the code has still not been provided which makes it difficult to verify the results. In addition, the authors haven't included any additional results with suggested baselines, ablations, and more challenging environments for a more thorough evaluation of the proposed method.
> >
> > Given these reasons, I cannot recommend acceptance at this point. I suggest to take the feedback into account and resubmit an improved version of the paper at a different venue.

---

### Official Review · Reviewer_5gPZ · 2022-10-24

**Confidence:** 2
**Correctness:** 2
**Technical Novelty And Significance:** 2
**Empirical Novelty And Significance:** 2
**Recommendation:** 3

**Clarity, Quality, Novelty And Reproducibility:**

address and the scope of their intentions for how their proposed method should be used and in what contexts.

I consider this paper to have low quality, using inappropriate experimental comparisons and including multiple inaccuracies.

I haven't come across any method like the proposed hippocampal replay (novel to the best of my knowledge) but my expertise is not around replay methods so I would not expect to be aware of similar methods. The use of a reward function that is -(distance to the target) is a fairly standard reward design, so I'm not sure what the novelty of GDIR is.

I haven't checked carefully for reproducibility because I think there are higher-level concerns. However, not being familiar with Nim, one question I had was, How do I determine what K is?

**Details Of Ethics Concerns:**

Figure 1 appears to be taken directly from the source, Joo & Frank (2018), which seems to be under Springer Nature copyright, as no mention of permissions is made. EDIT: The authors have explained that the figure is provided under a different license via PubMed, which I had not realized at review time!

**Strength And Weaknesses:**

The method described in the paper may have some interesting properties that should be further explored, but, unfortunately, I don't think the method was presented effectively. Most importantly, the choice of experiments is inappropriate. Providing the agent with a domain-specific distance function reward is providing it with domain knowledge, and so the comparisons against methods that are not provided this expert knowledge are not very meaningful.

However, I understand the position of having an interesting idea that still has some gaps in it, and see the merit in sharing such an idea with the community, even without being able to show off a complete system. In such a case, rather than trying to compare performance, I'd recommend experiments that help build intuition, perhaps using toy domains, and being extremely clear whenever the system designers have had to fill in a gap with a bit of a hack, like providing appropriate subtasks for the agent if it was beyond the scope of the paper to find a method that could do so autonomously. I thought the ablation study was a good idea (strength) and would fit really well into a framework of experiments aimed at demonstrating the potential for the proposed method.

Another weakness of the paper is in how the context of existing work is presented. Prior work is often listed or briefly summarized without any explanation of how it is related to the paper and the proposed methods (e.g. Chentanez et al. (2004), Schmidhuber (2010), Baldassarre & Mirolli (2013), Aubret et al. (2019; 2022), Yang et al. (2022), and Gallouédec & Dellandréa (2022); pp. 2-3). There are also several misleading statements about prior work in the authors' descriptions of prior work, like the description of first-return-then-explore as an extension of Go-Explore ("First return, then explore" is a paper presenting the Go-Explore family of algorithms, not an algorithm) and the statement that Oudeyer & Kaplan (2009) give a list of computational approaches to model intrinsic motivation in psychology (many of the listed reward functions, like FM, CM or StabM, while effectively demonstrating some of the possibilities of a computational analogue of intrinsic reward, are not meant to model anything that would be called intrinsic motivation in psychology).

Another strength of the paper is the inclusion of substantial detail about the implementations of the experiments. The authors have been fairly careful to include parameter settings and have provided more than one description of each experiment to help the reader understand.

Some questions/comments I would especially recommend addressing in revising this paper:

1. To the best of my understanding, this method is designed to work with some kind of subgoal-state generator, like hierarchical reinforcement learning (in reference to p. 2); do the experiments include this?
2. I found the use of the words "goal," "sub-task" and "task" quite confusing—I assume in most cases that these were references to the outputs of a subgoal-state generator like I mentioned in Question 1, but sometimes I wondered if the environment was also supposed to be formulated as having a single goal state? I'd recommend defining the term you use explicitly and using the same term consistently to reduce confusion.
3. "Recently, LeCun (2022) describes …" (p. 1) → I don't think I understand the point of your first paragraph. Removing it would likely reduce my confusion and save you some space.
4. There are multiple references to "curiosity-based" or "curiosity-driven" intrinsic motivation but no definition of what the term refers to. The term "curiosity" gets used for multiple things, so it is valuable to be clear about which ones are meant.
5. "but the extrinsic reward may not be obtained." (p. 2)→ I'm not sure what this means.
6. "For a new memory state," (p. 4) ← Not sure what a "memory state" is—are you just referring to when you store a state to memory for the first time?
7. "in a deterministic (rather than probabilistic) manner to help the algorithm learn faster." (p. 3) ← Can you provide an explanation for how deterministic versus probabilistic choice speeds up learning (even if it is only in the appendix)? It seems like there must be a reason the original authors used the probabilistic choice.

Typos and grammatical suggestions (no intended influence on score)

1. "Motivated by these observations, we seek to find such intrinsic cost/reward functions whereby it is innate to the agent, but is context dependent and can be triggered according to the task at hand." (p. 2) → There's some noun-verb disagreement here, so you could use something like, "Motivated by these observations, we seek intrinsic cost/reward functions that are innate to the agent, but context dependent and which can be triggered according to the task at hand."
2. "tells us what are the sub-tasks" (p. 2) → "tells us the sub-tasks
3. "Oudeyer & Kaplan (2009) describes" (p. 3) → "Oudeyer & Kaplan (2009) describe"
4. "Go-Explore." (p. 2) ← I recommend providing the reader with a summary/review the the "Go" and "Explore" phases of Go-Explore in preparation for Section 3.
5. "Go-Explore (Ecoffet et al., 2019) describes …" (p. 2) → Go-Explore is an RL algorithm (or family of algorithms) so it typically doesn't describe, though Ecoffet et al. do! Also, as a note, detachment and derailment are also described in the published version of the paper (see p. 580 and pp. 19-20 in the supplementary material) so it might be preferable to cite the Nature publication here.
6. "due to the in-built stochasticity of the algorithm" (p. 2) → I'd recommend foreshadowing how you'll address these problems here. This would also be a great place to provide some intuition about why most algorithms have in-built stochasticity.
7. "We seek to guide the search via a goal-directed approach, much like using a compass to find the way in the forest." (p. 3) ← While I appreciate your aim to build intuition, I don't understand the analogy. What makes using a compass goal-directed? The goal is probably not simply to use the compass, maybe it is to go in a particular direction?
8. "where it can be viewed" (p. 3) ← Where what can be viewed?
9. "which is ideal in sparse reward settings. … GDIR is innate to the agent and hence applicable across a wide range of environments without needing to rely on a properly shaped environment reward." (p. 3) ← There seems to be a contradiction here, because sparse reward settings obviously don't have potential-based shaped environmental reward; it would be helpful to have more intuition on how potential-based reward shaping is usually used.
10. "baseline for all other agents." (p. 3) ← I'd leave this at "serves as the worst-case baseline" because I'm not sure how other agents come into it.
11. "its superior performance in them." (p. 3) ← Not sure what "them" is in this sentence.
12. "Guillaume Chaslot, Sander Bakkes, Istvan Szita, and Pieter Spronck. Monte-carlo tree search: A new framework for game ai." (p. 10) → "Guillaume Chaslot, Sander Bakkes, Istvan Szita, and Pieter Spronck. Monte-Carlo tree search: A new framework for game AI."
13. "Marek Grzes. Reward shaping in episodic reinforcement learning. 2017." ← I think this is an AAMAS paper, but this could use more citation detail.
14."P Read Montague. Reinforcement learning: an introduction, by sutton, rs and barto, ag. Trends in cognitive sciences, 3(9):360, 1999." (p. 11)

**Summary Of The Paper:**

In this paper, the authors propose a method to speed up learning through the use of a reward function guiding the agent to complete subtasks. Additionally, they propose a replay method, which they call "hippocampal replay." The authors include an empirical demonstration of their method in both continuous and discrete environments, as well as an ablation study to explore the contributions of both the choice of reward function and the use of hippocampal replay.

**Summary Of The Review:**

As a proviso, one issue I had with this paper is clarity. While I tried hard to understand the paper and am reasonably well-versed with most intrinsic reward methods, I really struggled to understand what the authors were trying to communicate.

The primary reason I am recommending rejection for this paper is that, while the central contribution is supposed to be an intrinsic reward method, the authors don't actually use intrinsic rewards. They hand-design different reward functions as the "GDIR" for each test domain (Section 4.3), which is a key indicator that the rewards are not intrinsic. This means that their experiments comparing their methods with baselines don't provide any significant information; the authors compare their method, which is provided with an appropriate distance function as domain knowledge against methods that do not use domain knowledge. The paper states that the method isn't provided with domain knowledge (p. 4), but to the best of my understanding, such a statement isn't true.

---

> ### Author Response · Authors · 2022-11-18
> **Author Response**
>
> Thank you for the feedback. We will reframe this paper and focus more on discrete environments, but with various ablations to the method to try to gain insights on how and why it works. Initially, we were too focused to cover as many environments as possible, but I think for a good analysis, we will narrow down our scope to discrete environments first and seek to derive insights from it before the scale of our experiments.
>
> Addressing Ethics Concerns:
> The image is also available on PubMed website, which allows reproduction with a citation. https://pubmed.ncbi.nlm.nih.gov/30356103/
>
> Answers to Questions:
>
> 1.	The subgoal/goal generator is the overall context behind this GDIR. We believe that how the brain processes is that there is a brain region (likely the frontal lobe) which controls planning and goal generation, which then serves as a basis for evaluating actions thereafter. We treat this goal setting as given (this is a difficult problem which is best tackled separately), and try to see if given the goal, can the algorithm we propose solve it efficiently. We show that our method can solve the environments more efficiently in most cases (lower minimum number of steps to solve) with the inclusion of GDIR, which serves to prove our hypothesis.
>
> 2.	You are right to discern that the goal, sub-task and task are actually the same in this paper, as we are dealing only with the sub-task/sub-goal here. It is also interesting to note that our environments only have one goal. How will GDIR direct the agent if there are multiple goals remains an open question. Perhaps there could be an abstraction in the GDIR metric such that heading towards any of the possible goals would lead to an increased GDIR.
>
> 3.	We will make the introduction tighter as per your recommendation in the revised paper. We only included it here because Yann LeCun was the primary inspiration behind us trying to use intrinsic reward instead of the typical extrinsic/environment reward. However, merely trying to use existing intrinsic reward formats (i.e. curiosity-driven) is not ideal due to the problems of detachment and derailment as mentioned in the Go-Explore paper. Hence, we tried to derive our own version of intrinsic reward.
>
> 4.	The curiosity-driven reward we are referring to are the ones with some kind of novelty bonus added to a state if it is unvisited before. We will take note of this in our revised paper.
>
> 5.	This means that because the reward to explore a particular region may be so low, the actual high extrinsic reward (i.e. solving the environment in sparse reward settings) may not be obtained. Without this extrinsic reward, we are then unable to credit assign and learn the successful trajectory.
>
> 6.	The memory state refers to the attributes of a state that is stored. They are namely: 1) trajectory of actions to reach that state, 2) the number of moves to reach it, 3) the extrinsic reward, 4) the intrinsic reward (unique to our paper), 5) the number of selections of the state in the ”Go” phase, and 6) the number of visits to the state in the ”Explore” phase
>
> 7.	The deterministic choice means that we can reach the desired goal state in fewer runs. With the probabilistic version, there is a non-zero probability of choosing any state previously encountered, and this necessarily leads to inefficiencies in exploring the environment as some potentially bad states may still be visited. The original authors use a probabilistic method because their metric was not the number of runs it takes to solve the environment – in fact, they used in excess of a week to obtain the superhuman performance on Montezuma’s Revenge, which is a lot of computation necessary to get the solution. We also posit that the original authors use a probabilistic method just in case their heuristics did not choose the right state, and the non-zero probability would ensure that in the long run all states will be explored.
>
> What K is in Nim: K is the maximum number of matchsticks one can take out per turn in Nim. Generally, Nim starts off with N matchsticks (typically 21), and then players take turn to remove 1-3 (K=3 in this case) matchsticks from the pile. The player who removes the last matchstick wins.
>
> Thank you once again for the detailed feedback, which will be taken into account for our revised paper!

---

### Official Review · Reviewer_j1xa · 2022-10-25

**Confidence:** 4
**Correctness:** 3
**Technical Novelty And Significance:** 2
**Empirical Novelty And Significance:** 2
**Recommendation:** 3

**Clarity, Quality, Novelty And Reproducibility:**

- Clarity: Some claims remain a bit vague: e.g. in the introduction, the authors claim they aim for a satisfactory solution, rather than a perfect one - but what is being satisfied? It looks like a change of reward to me, without stating how the reward is changed. The proposed method is explained clearly enough, so I don’t have major concerns on this front. The paper would benefit from a round of language editing, see a few examples below. (I am not basing my recommendation on this, but wanted to point it out to the authors.)
  - Silver et al. (2021) … he admits -> they admit; there are other authors on that paper.
  - Satisficing: satisfying / satisfactory?
  - “we seek to find such intrinsic cost/reward functions whereby it is innate to the agent, but is context dependent and can be triggered according to the task at hand” - please rephrase; what is innate to the agent?
- Quality: The reliance on simulators and hand-crafted intrinsic rewards and preprocessing functions makes the proposed method unlikely to generalize well and have many practical benefits. The experimental evaluation does demonstrate that the idea can work across various domains, although showing results on more specific hard exploration tasks would make the paper stronger. The missing baselines mentioned above make it harder to assess the benefits of the method.
- Novelty: The individual elements in the paper may not be particularly novel, but the combination of intrinsic rewards and Go-Explore has not been used before as far as I'm aware. That is a good idea, but could be demonstrated more convincingly.
- Reproducibility: no concerns.

**Strength And Weaknesses:**

- Strength: the method is tested, and shown to generally outperform the chosen baselines, across multiple environment types and tasks.
- Strength: the core of the paper - the method - is explained clearly.
- Weakness: the GDIRs are hand-crafted for each task, which harms the generalizability of the method.
- Weakness: the Go-Explore aspect of the method relies heavily on being able to use simulators. The authors recognize this dependence, but it still reduces the applicability of the method in practice.
- Weakness: the pre-processing of different environment types is somewhat specific to the proposed method, in particular the Go-Explore aspect.
- Weakness: some baselines are missing, e.g. novelty signals based on learned representations, and non-Go-Explore agents with the GDIRs. Some citations are also missing, e.g. Random Network Distillation by Burda et al., Curiosity-driven Exploration by Self-supervised Prediction by Pathak et al., and Never Give Up / Agent57 by Badia et al.

**Summary Of The Paper:**

The paper proposes the use of goal-directed intrinsic rewards (GDIR), and evaluates them in a Go-Explore-based agent on a variety of discrete environments, as well as some continuous control ones. The goal-directed intrinsic rewards are hand-crafted for each task, and the authors choose functions that are similar to the goals many human players would perceive - e.g. Manhattan distance to a goal in a discrete maze. The GDIRs are only used in the Go-Explore-based agents, not in the baselines. Generally, the GDIR-enhanced agents perform better than the baselines, although the difference with respect to GDIR-less Go-Explore is not always equally pronounced.

**Summary Of The Review:**

The basic idea behind the paper, using intrinsic rewards in Go-Explore, is a good idea. However, the implementation presented here looks very brittle, due to heavy reliance on handcrafting and simulators. As it stands, the paper is not making a convincing case that the proposed method will generalize well. I recommend rejection.

---

> ### Author Response · Authors · 2022-11-17
> **Author Response**
>
> Thank you for your feedback. The satisficing solution (definition: a course of action that will satisfy the minimum requirements necessary to achieve a particular goal.) is actually inspired from real life. In real life, the agents do not have thousands of simulated time to adapt and learn from the environment, they actually only have one life and a very limited amount of time to gain experience. Hence, in contrast to majority of RL algorithms, we seek not to maximize the reward, but just to solve the environment and try to repeat this successful trajectory as much as possible using hippocampal replay. This will not lead to the most optimal solution, but at least the agent will be more survivable and can learn and remember successful solutions.
>
> Thank you for agreeing that GDIR is a great idea!! It is actually inspired from the fact that for humans, if we want to head straight towards a door from your chair in a room, we can actually go right towards it, but for RL agents, in general they have to do some form of exploration around the room and may just “stumble” upon the door. This is just one example of inefficiency current RL algorithms have and we seek to put GDIR as part of the intrinsic reward (not environment reward), because it can be generically applied across any environment as long as we can quantify the difference between the start state and the goal state.
>
> We agree that the intrinsic rewards as of now are hand-crafted for each environment, but in the future, we intend to generalize this using some form of generic distance metric such as cosine similarity between states. This will be explored in future work.
>
> We will also implement GDIR on the non-Go-Explore agents and compare their performance to provide a more complete benchmark.
>
> Also, another point of note – there is a method proposed, Explore-Count, which does not require “teleporting” to the Go state. This will not require simulator help (only requires a one-step lookahead internal model) and will be well-suited to how we learn in real life.

---

> > ### Comment · Reviewer_j1xa · 2022-12-08
> > **Unchanged**
> >
> > Apologies for my late response.
> >
> > I agree that data efficiency in RL is a worthwhile goal. I would maintain that defining a satisficing solution comes down to an alternative reward structure, unless the reward happens to be aligned with that solution to begin with - in which case optimal and satisficing solutions should not be very different.
> >
> > I look forward to the future work with less hand-crafted intrinsic rewards! I believe that is quite non-trivial, and essential for the whole method to generalize well. In the current form, the paper still does not provide convincing evidence of that, so my score is unchanged.

---

### Official Review · Reviewer_JKcR · 2022-10-26

**Confidence:** 3
**Clarity, Quality, Novelty And Reproducibility:** The paper is clearly written.
**Correctness:** 3
**Technical Novelty And Significance:** 2
**Empirical Novelty And Significance:** 2
**Recommendation:** 3

**Strength And Weaknesses:**

Strength:
1. The paper tackles an interesting problem of exploration in the sparse-reward RL environments.
2. The paper builds upon a strong baseline: Go-explore.
3. The paper compares thoroughly against multiple baselines.

Weakness:
1. My main concern with the paper is that the experiment mostly builds on toy tasks, e.g., 2D maze, Mountain Car, and Cart Pole. In addition, the paper didn't talk about the random seed for their experiments, which is a pretty standard evaluation protocol for modern RL methods. Lastly, the evaluation metric used in the paper is somewhat strange. Why not report the success rate?

**Summary Of The Paper:**

This paper talks about the exploration of sparse reward environments. Based on the Go-explore paper, the paper proposed a goal-conditioned potential-based intrinsic reward for goal-conditioned tasks. The paper conduct experiments on the 2D grid maze environments, Towers of Hanoi, Game of Nim, Mountain Car and Cart Pole.

**Summary Of The Review:**

Please see the strength and weaknesses above.

---

> ### Author Response · Authors · 2022-11-17
> **Author Response**
>
> Thank you for the feedback. We are currently working on more advanced algorithms like that of MuJoCo using policy gradient methods which incorporate GDIR. The toy examples are used for this paper because this is a novel idea and we want to test it out on small testbeds before scaling it up.
>
> The random seed used was from 1 to 100, across all 100 runs of the experiment. We will include this in the revised paper.
>
> The success rate is reported as “Solve Rate” in the appendices. It was removed in the main paper due to lack of space, but it could be added in for clarity. Methods with GDIR achieves a higher solve rate in general than methods without it.

---

### Decision · Program_Chairs · 2023-01-20

**Decision:**

Reject

**Justification For Why Not Higher Score:**

Reviewers were all agreed that this paper in its current form is not suitable for publication at ICLR, citing many reasons, the most significant of which is that their approach wouldn't generalize beyond these specific 5 tasks, since their intrinsic rewards are hand-crafted for each task and their method relies on access to simulators. Other reasons for not giving a higher score include the overall lack of clarity and contextualization within related work.

**Justification For Why Not Lower Score:**

N/A

**Metareview: Summary, Strengths And Weaknesses:**

This paper proposes an extension of Go-Explore, incorporating potential-based goal-directed intrinsic rewards (GDIFs) as well as replay of successful trajectories in order to deal with sparse-reward scenarios. They tested on 3 discrete tasks (2D grid maze environments, Towers of Hanoi, and Game of Nim) and 2 continuous control ones (Mountain Car and Cart Pole).

Reviewers agreed that the motivation is strong, namely the problem of exploration in sparse reward environments. Other strengths include good presentation of experimental detail (j1xa, 5gPZ) and a comparison to multiple existing baselines (JKcR). The ablation study was also commended by one reviewer (5gPZ).

However, several major noted weaknesses are the toy nature of the tasks (JKcR, eHGy), that the GDIRs needed to be handcrafted for each task (j1xa, 5gPZ, eHGy), reliance on simulators (j1xa, eHGy), and inadequacy of the baselines (j1xa, 5gPZ, eHGy) and explanation of related work (5gPZ, eHGy). This paper overall seems to lack proper contextualization within the already robust field of exploration in RL, and reviewers point to many missed references (eg Burda et al, Pathak et al, Badia et al).

Authors don’t seem to address the main critical points of reviews, namely to do with the hand-crafted nature of their intrinsic rewards, which make use of domain-specific knowledge. I also don't believe that this paper hasn’t been updated from the original submitted version. Therefore I unfortunately have to recommend reject, but I hope that these reviews will help the authors improve the paper for future submissions.